

# Is the photochemistry activity weak during haze events?
## —— A novel exploration on the photoinduced heterogeneous reaction of
## NO₂ on mineral dust
Tao Wang[1], Yangyang Liu[1], Yue Deng[1], Hanyun Cheng[1], Yang Yang[1], Yiqing Feng[1], Muhammad Ali Tahir[1],
Xiaozhong Fang[1], Xu Dong[1], Kejian Li[1], Saira Ajmal[1], Aziz-Ur-Rahim Bacha[1], Iqra Nabi[1], Hongbo Fu[1], Liwu
Zhang[1,2*], Jianmin Chen[1]
[1] Shanghai Key Laboratory of Atmospheric Particle Pollution and Prevention, Department of Environmental
Science & Engineering, Fudan University, Shanghai, 200433, Peoples' Republic of China
[2] Shanghai Institute of Pollution Control and Ecological Security, Shanghai, 200092, Peoples' Republic of China
**Abstract**
Despite the increased awareness of heterogeneous reaction on mineral dust, the knowledge of how the
intensity of solar irradiation influences the photochemistry activity remains a crucially important part in
atmospheric research. Relevant studies have not seriously discussed the photochemistry under weak sunlight
during haze, and thus ignored some underlying pollution and toxicity. Here, we investigated the heterogeneous
formation of nitrate and nitrite under various illumination conditions by laboratory experiments and field
observations. Observed by *in-situ* diffuse reflectance infrared Fourier transform spectroscopy (DRIFTS),
water-solvated nitrate was the main surface product, followed by other species varying with illumination condition.
The growth of nitrate formation rate tends to be slow after the initial fast with increasing light intensity. For
example, the geometric uptake coefficient ($\gamma_{geo}$) under 30.5 mW/cm² ($5.72 \times 10^{-6}$) has exceeded the 50 % of that
under 160 mW/cm² ($1.13 \times 10^{-5}$). This case can be explained by the excess NO₂ adsorption under weak illumination
while the excess photoinduced active species under strong irradiation. Being negatively associated with nitrate
($R^2=0.748$, $P<0.01$), nitrite acts as the intermediate and decreases with increasing light intensity via oxidation
pathways. Similar negative dependence appears in coarse particles collected during daytime ($R^2=0.834$, $P<0.05$),
accompanied by the positive association during nighttime ($R^2=0.632$, $P<0.05$), suggesting illumination a
substantial role in atmospheric nitrogen cycling. Overall, for the nitrate formation, the conspicuous response under
slight illumination offers opportunities to explain the secondary aerosol burst during haze episodes with weak
irradiation. Additionally, high nitrite levels accompanied by low nitrate concentrations may induce great health
risk which was previously neglected. Further, Monte Carlo simulation coupled with sensitivity analysis may
provide a new insight in the estimations of kinetics parameters for atmospheric modelling studies.


## 1 Introduction

Secondary nitrate aerosols, deriving mainly from various oxidation processes of nitrogen oxides, are of great importance in atmospheric chemistry (**Anenberg et al., 2017**). These ubiquitous species is key for describing the composition and sources of particulate matters (**Huang et al., 2014; Schuttlefield et al., 2008**). It was investigated that the contributions of nitrate to the particle mass concentration increase throughout the pollution episodes (**Guo et al., 2014**). However, current atmospheric models fail to capture the serve nitrate enhancement from the clean to haze period, and thus triggers the discussion on the heterogeneous reaction of $NO_2$ on primary aerosols (**Tan et al., 2016**). Modeling studies indicated that nitrate formation is highly associated with airborne mineral dust (**Tan et al., 2016**). Accounting for ~36% of the total primary aerosol emissions, mineral dust is one of the most abundant particle types in the troposphere (**Chen et al., 2012; Usher et al., 2003**). During their global journey, many heterogeneous reactions take place on the particle surface, and further affect the atmospheric oxidation capacity (**Tang et al., 2017**). This process has aroused widespread interest in East Asia because dust occupies a great share in fine particles due to the frequent occurrence of sand storms (**Zhang et al., 2015**). Hence, the heterogeneous reaction of $NO_2$ on mineral dust is worthy of broader concerns.

Titanium dioxide ($TiO_2$) is found in mineral dust at mass mixing ratios ranging from 0.1% to 10% depending on the exact location from where the particles were uplifted (**Ndour et al., 2008**). Compared to other non-semiconducting components in mineral dusts, $TiO_2$ has direct environmental implications for its photocatalysis (**Nanayakkara et al., 2014**). Prior studies have indicated the photoinduced oxidation of trace gases by $TiO_2$ an essential role in the chemical balance of the atmosphere (**Chen et al., 2012**). Additionally, $TiO_2$-coating surfaces are currently used on building exteriors, road lamps and road bricks (**Ballari et al., 2010; Ballari et al., 2011**). These self-cleaning materials in populated urban areas facilitate the irreversible removal of $NO_2$ from atmosphere with the substantial formation of gaseous nitrous acid (HONO) and ozone (**Balajka et al., 2018; Langridge et al., 2009; Monge et al., 2010**). Accordingly, $TiO_2$ is frequently adopted as the reference material on behalf of the ubiquitous semiconducting components in atmospheric environment, especially the urban atmosphere.

For the heterogeneous process on mineral dust, prior studies put close attention to varied influential factors. Among these, moisture and temperature are widely concerned and significant advances have been made (**Li et al., 2010; Tan et al., 2017; Tan et al., 2016; Wang et al., 2012**). Although being treated as an important index in many atmospheric discussions, illumination has not been systematically investigated for its effects on the heterogeneous uptake of trace gases. Most remarkable studies concerned the photocatalytic effects instead of the



dependence on illumination conditions **(Dupart et al., 2014; Guan et al., 2014)**. Some researchers **(El Zein and**
**Bedjanian, 2012)** measured the reactive uptake coefficients (γ-values) for the heterogeneous reaction of $NO_2$ on
$TiO_2$ under various irradiance intensities while ignored the reaction mechanism behind the variation. Furthermore,
nitrite is of great significance in atmospheric processes for its frequent appearance and great contributions to
aerosol toxicity. However, there is little information available in literature about the pollution characteristics or
reaction pathways of nitrite aerosols. Generally, how the illumination influences the uptake capacity and product
species are problems urgently needs solving.

67        This work aims to provide a fresh perspective to explore the light dependence for the heterogeneous reaction

on mineral dust. Monte Carlo simulation is introduced to evaluate the kinetics for nitrate formation. Atmospheric
particulates were collected and analyzed to support relevant findings. This research could help further understand
the illumination effects in the atmospheric nitrogen cycling, and simultaneously provide extremely valid
parameters for modelling studies.
**2    Experimental**
**2.1   Materials**

74        Commercial $TiO_2$ (Degussa, Germany), with an anatase-to-rutile ratio of 3: 1, an average particle size of

12.05±3.46 nm and a Brunauer-Emmett-Teller (BET) specific surface area ($S_{BET}$) of 55.83±0.35 $m^2 \cdot g^{-1}$ was
employed as the photocatalytic mineral dust *(SI, Section S1)*. All chemicals were of analytical grade and obtained
from Aladdin Chemical Reagent Co., Ltd. Water in all experiments was ultrapure water (specific resistance ≧ 18.2
MΩ cm).

79        High-pure air (79% $N_2$ and 21% $O_2$, Shanghai TOMOE Co., LTD, China) and 300 parts per million (ppm)

$NO_2$ ($N_2$ dilution, Shanghai Qingkuan Co., LTD, China) were included in this research. Prior to coming into the
gas supply system, high-pure air went through silica gel and molecular sieve for drying and purification.
**2.2   DRIFTS experiments**

83        A FTIR spectrometer (Tracer-100, Shimadzu, Japan) equipped with a liquid-nitrogen-cooled

mercury-cadmium-telluride (MCT) detector was applied to record in situ DRIFTS spectra with 100 scans averaged
for each spectrum and a resolution of 4 $cm^{-1}$. We have described the general features of the setup in *Figure S2* and
previous reports **(Wang et al., 2018**a**; Wang et al., 2018**b**; Wang et al., 2018**c**)**. Herein, a xenon lamp
(CEL-TCX250, Beijing Ceaulight Co., LTD, China) was used to provide simulated solar irradiation upon the
particles *(Figure S3)*.



89 Prior to each experiment, the particles were pretreated in a stream of high-pure air (200 ml·min⁻¹) for 60 min

90 to remove the adsorbed water and impurities from the surfaces *(Figure S4)*. Due to the overlapping bands of

91 adsorbed water (~1640 cm⁻¹) and nitrogen compounds, the sample after pretreatment was exposed to humid

92 high-pure air (RH≈30%, 100 ml·min⁻¹) for 20 min, after which the moisture absorption reaches saturation *(Figure*

93 *S5)*. A background spectrum was recorded after the process and then NO₂ calibration gas (5.12 ml·min⁻¹) was

94 added into the DRIFTS chamber with a calculated concentration of 15.33 ppm. Calibration gases with NO₂

95 concentrations of 9.20 and 21.45 ppm were also involved for the concentration dependence experiments. Ten

96 light intensity levels (0.0, 0.3, 5.4, 17.5, 23.8, 30.5, 54.5, 98.5, 128.1, and 160.0 mW·cm⁻²) were referred in this

97 study.

98 Each test lasted 90 min, during which a series of spectra were recorded every 5 min. The reacted particles

99 were extracted by oscillation (5 min) with 4 ml water. The extraction solution was then passed through a 0.22 μm

100 PTFE membrane filter for ion detection.

101 **2.3 Ion analysis**

102 The nitrate and nitrite ions were analyzed by an ion chromatography (IC, 883 Basic, Metrohm, Switzerland),

103 which consists of an analytical column (A5-250) and a guard column. The detection was conducted by using 3.2

104 mmol·L⁻¹ Na₂CO₃ and 1.0 mmol·L⁻¹ NaHCO₃ at a stable flow rate of 0.70 ml·min⁻¹. Multipoint calibrations were

105 performed by means of standard solutions. Good linearity of the calibration curve was obtained with $R^2$>0.998.

106 **2.4 Photo-electrochemical (PEC) test**

107 In order to qualitatively evaluate the generation of electron-hole pairs under different light intensities, PEC

108 tests were conducted by a electrochemical workstation (CHI-660D, Shanghai Chenhua Co., LTD, China) in a

109 three-electrode cell with a quartz window **(Yang et al., 2017; Zheng et al., 2015)**. TiO₂ particles were deposited

110 on a sheet of fluorine-tin-oxide glass to serve as the working electrode with an effective area of 1cm². A platinum

111 wire and an Ag/AgCl electrode were employed as the counter and reference electrodes, respectively. The

112 electrolyte was 0.5 mol/L NaNO₃. A xenon lamp (CEL-S500, Beijing Ceaulight Co., LTD, China) was used to

113 provide simulated sunlight.

114 **2.5 Uptake coefficient estimation**

115 The reactive uptake coefficient, γ, is defined as the ratio of the reactive gas-surface collision rate ($\mathrm{d}[NO_3^-]/dt$)

116 to the total gas-surface collision rate (Z) **(Gustafsson et al., 2006)**. The equations are shown as follows.

117 $$\gamma = \frac{d[NO_3^-]/dt}{Z} \qquad\qquad Eq.(1)$$





$$\frac{d[NO_3^-]}{dt} = slope \times f \qquad Eq.(2)$$


$$Z = \frac{1}{4} \times A_S \times [NO_2] \times v_{NO_2} \qquad Eq.(3)$$


$$v_{NO_2} = \sqrt{\frac{8RT}{\pi M_{NO_2}}} \qquad Eq.(4)$$


Where *slope* represents the growth rate of the nitrate peaks, *f* is the conversion factor, $A_s$ is the particle
reactive surface area, $v_{NO_2}$ is the mean velocity of NO2 molecule, [NO2] is the NO2 concentration, *R* is the gas
constant, T is the temperature, $M_{NO_2}$ is molecular weight of NO2 *(Table S1)*.
The conversion factor (*f*) is obtained from a calibration plot with the amount of $NO_3^-$ versus the integrated
areas for nitrate    **(Tan et al., 2017; Tan et al., 2016)**. The factor is $2.09 \times 10^{15} \pm 1.61 \times 10^{14}$ (ion · K-M unit$^{-1}$) in this
study *(Figure S6)*. For the As, both geometric surface area (Ageo) and BET surface area (ABET) are mentioned to
evaluate the upper and lower limits of the γ-values (denoted as γgeo and γBET, respectively) varying with reaction
probabilities between reactants and particles.
Monte Carlo simulation was implemented to deal with the uncertainties **(Chiang et al., 2009; Ginsberg and**
**Belleggia, 2017; Xia et al., 2013)**. Each independent variable was determined via five or more replication
measurements and assumed to be normally distributed in the simulation. Based on earlier finds, 5000 iterations are
sufficient to ensure the stability of the results. Additionally, sensitivity analysis is helpful in exploring the variables
that influence the estimation most. Pearson correlation coefficients between each variable and the output (γ-value)
were calculated and then normalized to 100%. On this basis, the contribution of each input variable to the output
can be assessed. Three input variables are included for γgeo: *slope*, *f*, and $A_s$. For γBET, the As is further divided into
mass and SBET as discussed above.
**2.6 Particle sampling and chemical analysis**
Aerosols were collected in the late summer and early autumn in the campus of Fudan University, Shanghai,
China *(Figure S8)*. The first stage from 23th August to 17th September contains 26 daily samples. The second stage
lasted from 21th to 29th September, including eight sample sets collected during daytime and another eight during
nighttime.
The size-segregated samples ranging from 0.4 to 100 μm were collected on quartz fiber filters (Whatman, UK)
using an eight stage micro-orifice uniform deposit impactor (Anderson, Tisch Environmental Inc, USA) operating
at a flow rate of 28.3 L/min. The particle modes were defined as follows: 0-0.56 μm for condensation mode,
0.56-1.8 μm for droplet mode, and 1.8-100 μm for coarse mode.





Before sampling, the filters were pre-combusted at 550℃ for 4 h to minimize original impurities. After
collection, the filters were extracted ultrasonically by 20 ml water for 45 min. Water extracts were passed through
a 0.22 μm PTFE membrane filter for $NO_3^-$ and $NO_2^-$ detection as introduced in **section 2.3**.
**3    Results and discussion**
**3.1    Observed species on particles**
**Figure 1** presents the DRIFTS spectra recorded in the absence and presence of illumination, coupled with the
Gaussian curve-fitting procedure to deconvolute the overlapping bands. The fitting was undertaken until
reproducible results were obtained with the coefficient of determination ($R^2$) greater than 0.990. The bands in the
spectra are quite rich, indicating various products as summarized in *Table S3*.
Under illumination, the signals peaking at 1312 and 1553 cm$^{-1}$ reflect the formation of monodentate nitrate,
whereas those at 1276, 1573, and 1602 cm$^{-1}$ account for the vibration of bidentate nitrate **(Figure 1d, e) (Li et al.,**
**2010; Ma et al., 2011; Niu et al., 2017; Szanyi et al., 2007)**. Bridging bidentate nitrate can be further identified
by the shoulder peak at 1602 cm$^{-1}$ **(Du et al., 2019; Goodman et al., 1998; Sun et al., 2016)**. Besides, the peaks
at 1347 and 1412 cm$^{-1}$ are assigned to water-solvated nitrate **(Baltrusaitis et al., 2007; Guan et al., 2014; Miller**
**and Grassian, 1998)**. Under dark condition, except the similar bands appearing under illumination (1561, 1409,
1323, and 1271 cm$^{-1}$), some nitrite products become more attractive as evident by the monodentate nitrite at 1195
and 1440 cm$^{-1}$, as well as the bidentate nitrite around 1308 cm$^{-1}$ **(Figure 1a, b) (Wu et al., 2013)**. Water-solvated
nitrate is far ahead in amount compared to other species **(Figure 1c, f)**, suggesting weak links between the products
and particle surfaces. Hence, the surface water layers of the hygroscopic particles provides plenty active space for
the heterogeneous uptake of NO$_2$.
After the reversible adsorption of NO$_2$ on mineral dust **(R.S1)**, the NO$_2$ reacts with hydroxyl-related groups
(OH$^-$) or surface H$_2$O to form adsorbed nitrate/nitrite or free nitric acid/nitrous acid, respectively **(R.S2-S3)**. Since
no acid molecules were observed, free nitrite and nitrate ions stem from ionization **(R.S4-S5)**. The
disproportionation process **(R.1)** dominates the dark reaction. When excited with light (wavelength≥390 nm),
there is the generation of electron-hole pairs in the conduction and valence bands of TiO$_2$ **(R.S6) (Dupart et al.,**
**2014; FUJISHIMA and HONDA, 1972; Yu and Jang, 2018)**. Photogenerated holes and electrons react with
H$_2$O and O$_2$, and thus lead to the formation of hydroxyl radicals ($OH^{\cdot}$) and reactive oxygen radicals ($O_2^-$),
respectively **(R.S7-S8) (Chen et al., 2012)**. Superoxide hydrogen radical ($HO_2^{\cdot}$) and hydrogen peroxide (H$_2$O$_2$)
appear and produce $OH^{\cdot}$ as well **(R.S7-S14)**. These photoinduced active species (PAS) would accelerate the
nitrate formation **(R.2)**.





$$NO_2 \xrightarrow{OH^-/H_2O} NO_3^- + NO_2^-$$
R.(1)

$$NO_2 \xrightarrow{hv\ \&\ TiO_2} NO_3^-$$
R.(2)

Noticeably, nitrite (especially monodentate type) decreases in proportion as the dark reaction proceeds,
accompanied by the increasing contribution from bidentate nitrate species and water-solvated ones (*Figure 1c,*
*S10a*). The nitrite would react with another surface nitrite in a Langmuir-Hinshelwood mechanism (*R.3*) or
gaseous NO₂ in an Eley-Rideal mechanism (*R.4*) to form nitrate in the absence of illumination (**Tang et al., 2018;**
**Underwood et al., 1999**). Oxygen also acts as a promoter in the nitrite oxidation (**Tang et al., 2018**). On the other
hand, diverse nitrate species make steady contributions to the total products during the photoreactions (*Figure*
*S10b*). Generally, nitrite signal is visible in dark, while gradually fades away after irradiation due to the oxidation
of nitrite to nitrate by PAS via *R.5* (*Section S8*).
$$2NO_2^- \rightarrow NO_3^- + NO + e^-$$
R.(3)

$$NO_2^- + NO_2 \rightarrow NO_3^- + NO$$
R.(4)

$$NO_2^- \xrightarrow{hv\ \&\ TiO_2} NO_3^-$$
R.(5)

Illumination has impacts on either product species or the production. The final DRIFTS spectra grow in
intensity as the illumination becomes stronger. Raman measurements also indicate the drastic enhancement caused
by sunlight, evident by the higher nitrate peak after illumination compared to that after dark process (*Section S9*)
(**Fu et al., 2017; Yu et al., 2018; Zhao et al., 2018**). These observations provide a solid evidence that the nitrate
formation on mineral dust is enhanced under sunlight, in nice agreement with previous results (**Dupart et al.,**
**2014; Guan et al., 2014**). Noticeably, the nitrate determined by IC exhibits a clear nonlinear uptrend with
increasing light intensity, suggesting uneven illumination effect on nitrate formation (**Figure 1g**). On the contrary,
the nitrite presents a nonlinear downtrend (**Figure 1h**), and thus results in the negative association with nitrate
(**Figure 1i**). What is the proposed mechanism behind the uneven illumination effects? Whether the photoinduced
negative dependence appears in atmospheric particulates? We may discuss these issues in the following sections.



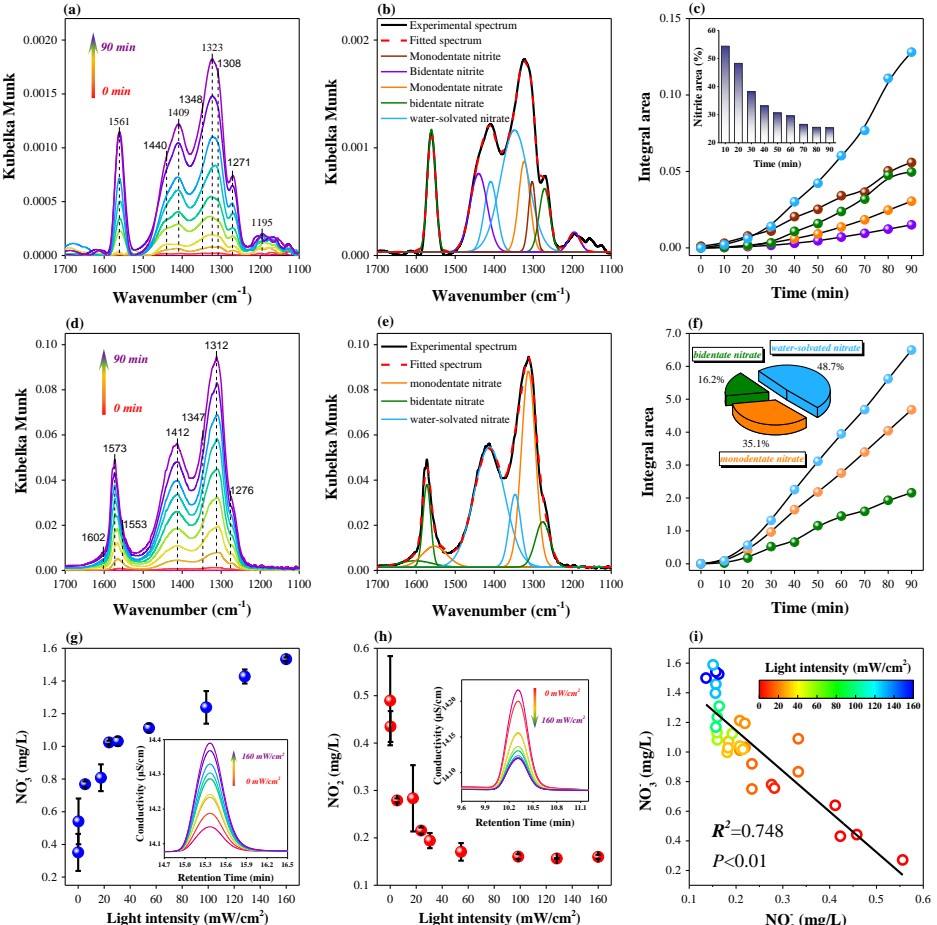


**Figure 1.** Product observations under **(a-c)** dark condition and **(d-f)** illumination (I=98.5 mW/cm$^2$), as well as **(g-i)**
ion analysis results. **(a, d)** DRIFTS spectra of nitrate and nitrate species. **(b, e)** Peak fitting for the final spectra
based on Gaussian method. **(c, f)** Integral areas of diverse species as a function of reaction time. IC measurements
for **(g)** nitrate and **(h)** nitrite ions after DRIFTS tests. Error bars represent 1σ. **(i)** Linear association between nitrate
and nitrite varying with light intensity. Inset: **(c)** Time-dependent contributions of nitrite, **(f)** Contributions of

205                  diverse products after 90 min exposure. **(g, h)** Conductivity spectra from IC.

**3.2  Photoinduced uptake capacity**

207         To accurately evaluate the photoinduced nitrate formation, γ-values were estimated based on Monte Carlo

simulation with the cumulative probability distributions depicted in **Figure 2** and the percentile values
summarized in *Table S2*. $\gamma_{BET}$ and $\gamma_{geo}$ exhibit similar variation trends. Since the reaction is first order with respect



to $NO_2$ concentration under various light intensities *(Figure S7)*, the γ-values would still be authentic for
atmospheric reactions with lower $NO_2$ concentrations. The growth of γ-values appears to be slow after the initial
fast with increasing light intensity. For instance, the $γ_{geo}$ under 30.5 mW/cm$^2$ exceeds the half of that under 160
mW/cm$^2$. To facilitate comparison, theoretical γ-values were calculated in a linear way based on the results under
0 and 160 mW·cm$^{-2}$. The actual γ-values under 5.4, 17.5, 23.8, 30.5, 54.5, 98.5, and 128.1 mW·cm$^{-2}$ are 73%,
135%, 189%, 158%, 148%, 103%, 39%, and 16% higher than the corresponding theoretical ones, respectively.
This 'fast-slow' uptrend seems to be of great importance as it shows that the γ-values measured at designed
irradiation intensity may not be extrapolated in a linear way to those relevant to the atmosphere. The balance
between PAS formation and $NO_2$ adsorption is responsible for the uneven illumination effect, which will be
carefully discussed in the mechanism section.
To distinguish the contributions of each variable to the output, sensitivity analysis is performed on the basis of
the simulated data. Slope and *f* contribute most to the total variance of $γ_{BET}$ and $γ_{geo}$, while $S_{BET}$ and m for $γ_{BET}$, and
$A_{geo}$ for $γ_{geo}$ contribute little *(Section S10)*. Accordingly, slope and *f* values in a more accurate level are beneficial
for γ-value estimation. More attention in the future needs to be devoted to the stability of DRIFTS and IC
measurements.
In view of the great significance of $γ_{geo}$ in atmospheric models, regression analysis is employed to fit the
obtained results and further predict values for relevant reactions. Since the γ-values exhibit a 'fast-slow' uptrend, a
polynomial regression model *(Eq.5)* is used to describe the variation.
$$γ_{f,geo} = aI^3 + bI^2 + cI + d \qquad\qquad Eq.(5)$$
Where $γ_{f,geo}$ is the fitted $γ_{geo}$, I is light intensity, and a, b, c and d are essential parameters. The final formula
*(Eq.6)* could explain 99.8% variation of the experimental $γ_{geo}$, indicating accurate regression **(Figure 2c)**.
Furthermore, the $dγ_{geo}/dI$ values are obtained by derivation to distinguish the illumination effect varying with light
intensity. The uptake capacity is extremely sensitive to light under low intensity, while tends to be
light-independent under strong irradiation **(Figure 2d)**. In Shanghai, the 3h-average intensities are mostly lower
than 80 mW/cm$^2$ (NOAA data, https://www.arl.noaa.gov/), indicating noticeable sunlight impacts. More
importantly, the irradiation tends to be weaker in winter, highlighting the central role of light-dependent
heterogeneous reaction in haze events during cold time.
$$γ_{f,geo} = 5.62 \times 10^{-12} \times I^3 - 1.92 \times 10^{-9} \times I^2 + 2.32 \times 10^{-7} \times I + 2.93 \times 10^{-7} \qquad Eq.(6)$$
$$dγ_{geo}/dI = 1.686 \times 10^{-11} \times I^2 - 3.84 \times 10^{-9} \times I + 2.32 \times 10^{-7} \qquad\qquad Eq.(7)$$



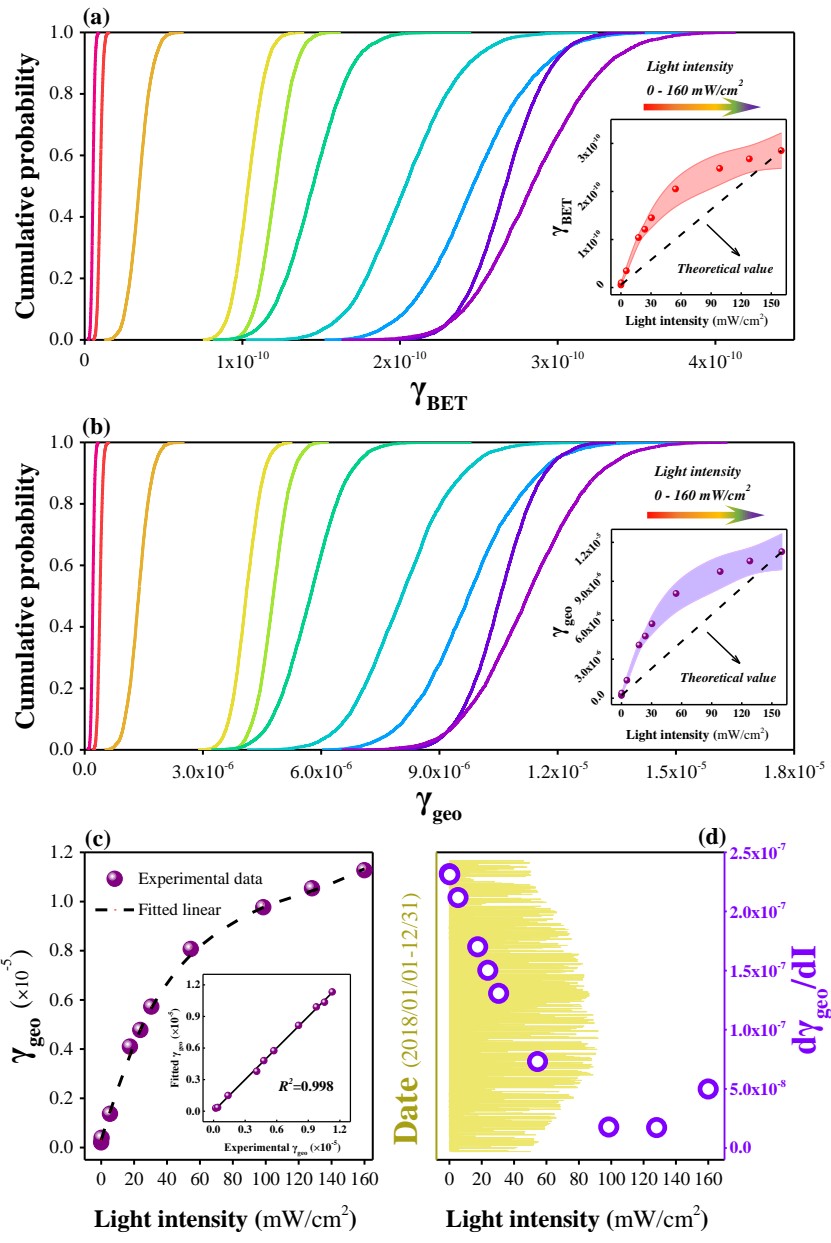


**Figure 2**. Cumulative probability distribution of the **(a)** $\gamma_{BET}$ and **(b)** $\gamma_{geo}$ values based on Monte Carlo

simulation. Insets reveal the actual γ-values (Mean±1σ) and theoretical ones. **(c)** Regression analysis on $\gamma_{geo}$.

Inset presents the linear relation between calculated and fitted values. **(d)** Downward short wave radiation

flux (DSWF) in Shanghai, China coupled with estimated $d\gamma_{geo}/dI$.



### 3.3 Nitrogen redox

**Figure 3** presents the association between atmospheric nitrate and nitrite varying with particle mode and sampling period. Significant positive correlation can be found during nighttime in coarse mode **(Figure 3a)**. However, there is no case indicating high nitrite and nitrate levels during daytime, and the dependence seems to be negative **(Figure 3b)**. The correlation turns to be significant with the ignorance of cases where the nitrite and nitrate concentrations are extremely low. As summarized in **section 3.1**, the associations during nighttime and daytime can be explained by the $NO_2$ disproportionation in the absence of sunlight **(R.1)** and the nitrite oxidation under illumination **(R.5)**, respectively. Daily nitrite and nitrate concentrations exhibit the similar variation with that during daytime, and a negative correlation can be observed based on the classification of nitrite levels **(Figure 3c)**. Noticeably, low nitrite levels are usually accompanied by slight nitrate pollution in the presence of sunlight, resulting mainly from the HONO formation in acidic media **(R.6)** **(Liu et al., 2015; Su et al., 2011; Wang et al., 2015; Zhang et al., 2012)** and the photolysis of particulate nitrate **(R.7)** **(Nanayakkara et al., 2014; Ostaszewski et al., 2018; Schuttlefield et al., 2008; Ye et al., 2017; Ye et al., 2016)**.

$$NO_2^- + H^+ \leftrightarrows HNO_2 \leftrightarrows HONO \qquad R.(6)$$

$$HNO_3 \overset{h\nu}{\Rightarrow} HONO + NO_x \qquad R.(7)$$

Atmospheric nitrate and nitrite from diverse periods exhibit analogous size distribution: greatest in coarse mode, followed by droplet mode and condensation mode **(*Figure S9*)**. Yet, except the large mass fraction in coarse mode, nitrite presents extra peak under 1.8 μm, indicating reaction pathways differing from nitrate formation **(Moore et al., 2004)**. That is, nitrate is difficult to accumulate by aqueous reactions or homogeneous processes while nitrite seems to be easy, which results in the lower correlation coefficients for small size particles **(Figure 3d-i)**. Since the main reaction pathways **(R.1, 3, 4)** still take place in aqueous media, and some other oxidants (e.g. $H_2O_2$, $O_3$, and $Fe^{3+}$) would replace the promoting role of semiconductor components in mineral dust under illumination **(R.2, 5)** **(Hems et al., 2017; Hou et al., 2017; Xue et al., 2016)**, the correlation in droplet mode appears to be obvious with merely lower coefficients. Furthermore, both ions exhibit great mass fractions (>50%) in coarse mode, making the associations for full-size particles similar with those for coarse aerosols **(Figure 3j-l)**.

Generally, atmospheric nitrite is positively correlated with nitrate during nighttime, whereas presents negative association with nitrate in the presence of irradiation. The dependence is significant in coarse mode while turns to be inconspicuous in droplet mode and condensation mode.

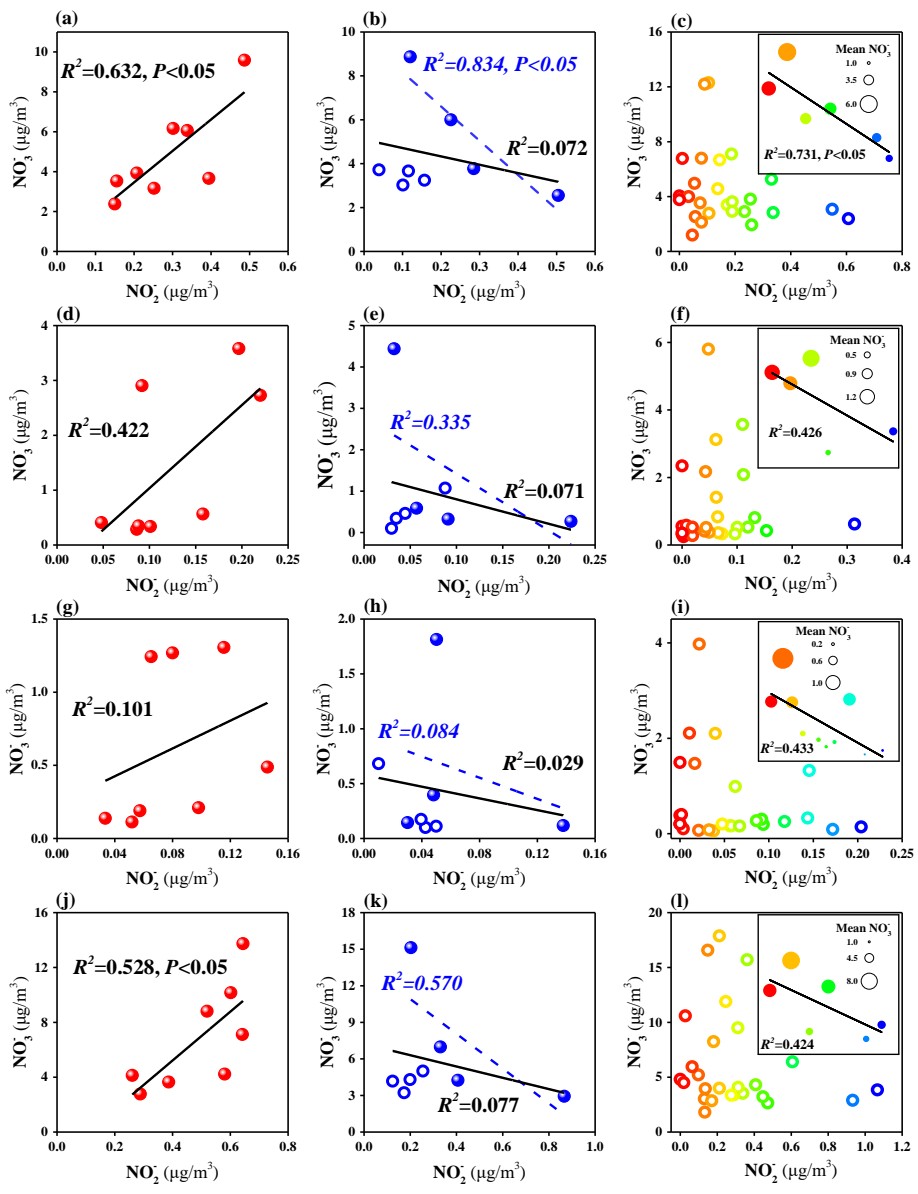

272

**Figure 3**. Associations between atmospheric nitrite and nitrate ions in **(a-c)** coarse mode (1.8-100 μm), **(d-f)**

droplet mode (0.56-1.8 μm), **(g-i)** condensation mode (0-0.56 μm), and **(j-l)** full-size particles (0-100 μm)

collected during **(a, d, g, j)** nighttime, **(b, e, h, k)** daytime, and **(c, f, i, l)** whole day. Insets: mean nitrate

concentrations based on nitrite classification. Linear correlation analysis (solid line) was employed for each case.

The dashed lines for daytime cases reveal the correlation for solid circles with hollow circles ignored.



### 3.4 Mechanism

Main reaction pathways have been introduced in previous sections. However, the uneven illumination effect cannot be explained by chemical equations. The photocurrent of $TiO_2$ is linearly correlated with light intensity **(Figure 4a)**, indicating even illumination effect on the production of electron-hole pairs. Hence, photocatalytic activity of the mineral dust is not sufficient to explain the uneven nitrate/nitrite formation with illumination variation.

Since no saturation effects were observed in the DRIFTS experiments, the $NO_2$ adsorption rate can be regarded as constant. Adsorbed $NO_2$ becomes excess compared to the formed PAS under weak sunlight, and thus makes illumination the rate-limiting factor in oxidation. At this time, nearly all the PAS participate in the oxidation of surface adsorbed $NO_2$ as well as some nitrite intermediates. When the illumination is strong, the PAS gradually become excess compared to the adsorbed $NO_2$. Under the circumstances, light makes little contribution to the elevation of uptake capacity and simultaneously $NO_2$ adsorption turns into the new rate-limiting factor. Generally, it could be deduced that the balance between $NO_2$ adsorption and PAS formation results in the nonlinear uptrend of γ-values with increasing light intensity.

Concentration dependence was considered in this research, and the nitrate formation rates at given conditions were normalized by the corresponding result estimated at 160 mW/cm$^2$ **(Figure 4b)**. At low concentration (9.20 ppm), the formation rate is light-dependent under weak illumination while tends to be steady with increasing light intensity, suggesting excess PAS under strong irradiation. The difference is that, at high concentration (21.45 ppm) the formation rate under strong irradiation was not nearly equal to that under 160 mW/cm$^2$, implying sufficient adsorbed $NO_2$ at relatively high intensity. Generally, higher $NO_2$ concentrations suggest broader influence scope of illumination. Hence, current serious $NO_2$ pollution may increase the participation of solar irradiation in the formation of secondary aerosols.



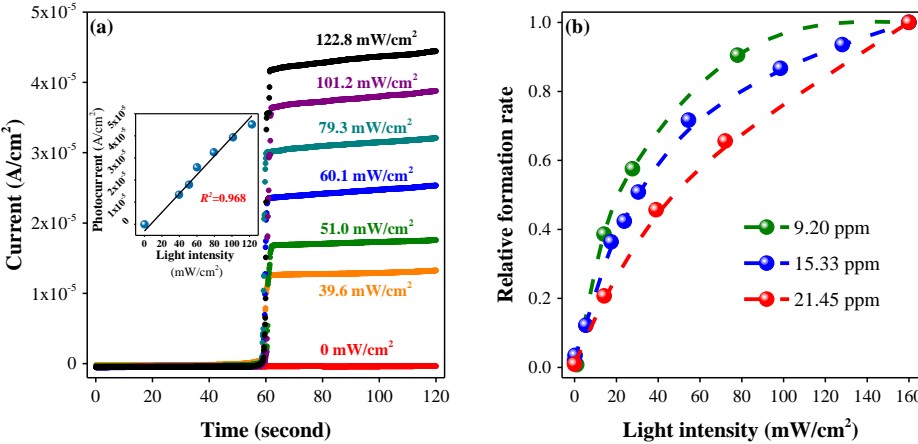


**Figure 4**. **(a)** Current densities of $TiO_2$ under various light intensities (60-120 s). Inset: linear correlation between
averaged photocurrent densities and irradiation intensities. **(b)** Relative formation rate of nitrate as a function of
light intensity under different $NO_2$ concentrations.
On the other hand, limited PAS participate in the oxidation of excess $NO_2$ under weak illumination, and thus
make space for the nitrite formation via disproportionation process **(R.1)**. Under strong irradiation, there are still
sufficient PAS involved in the nitrite oxidation after the photochemical conversion of limited $NO_2$ **(R.2)**. For the
oxidation of nitrite intermediates, the main promoters are $NO_2$ and other nitrite species under dark condition or
weak illumination **(R.3, 4)**, while turns to be PAS under stronger irradiation **(R.5)**. Hence, nitrite products
unevenly decrease with increasing light intensity.
As shown by **Scheme 1**, sunlight influences the formation pathways of nitrate and nitrite aerosols, and finally
results in reactions with different features. Mineral dust under weak illumination (or dark condition) and strong
irradiation may be covered by different nitrogen compounds: nitrite and nitrate, respectively. Since nitrite may
induce cancer risk while nitrate is generally treated as secondary pollutants, we can name them carcinogenic
aerosols and polluted aerosols in atmospheric research, respectively.



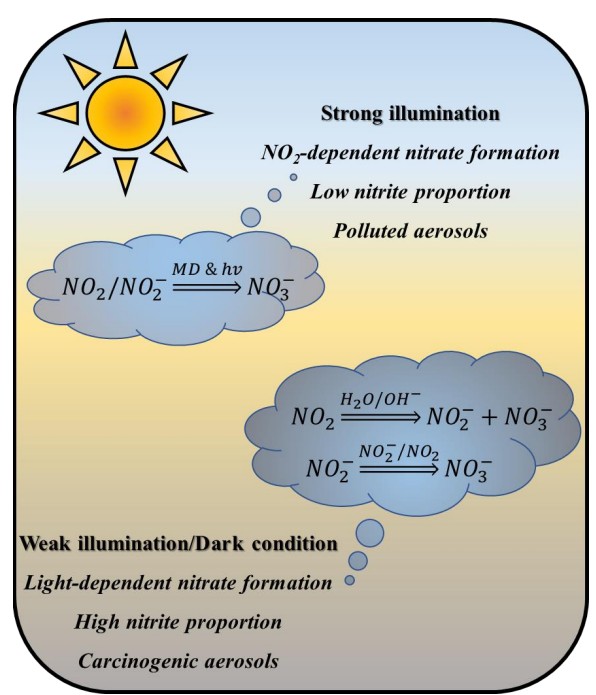


**Scheme 1**. Characteristics of the photoinduced heterogeneous reaction of $NO_2$ on mineral dust (MD) under

different illumination conditions.

## 4    Conclusions and atmospheric implications

Nitrate is dominating atmospheric particulates with the increasing $NO_x$ emissions from expanding urban

traffic (**Anenberg et al., 2017**). Photochemistry has traditionally been considered inapparent during haze events

because of the weak sunlight near the ground caused by low visibility (**Cheng et al., 2016; Shen et al., 2015;**

**Zhang et al., 2015**). However, the nitrate formation on mineral dust is found to be more dependent on weak

sunlight, indicating that photochemistry processes are still crucial in heavy haze. Since the $NO_2$ concentrations in

the troposphere are much lower than the simulated levels, authentic dust may be close to achieving its highest

uptake capacity in the presence of faint sunlight (**Figure 4b**). Hence, photoinduced reaction on mineral dust may

contribute greatly to secondary aerosols during extreme haze events.

Nitrate pollution has got much concern recently, while little attention has been paid to the nitrite burst

accompanied by low nitrate concentration. Nitrite may induce adverse health risk for its close association with

various cancer cases (**Zhang et al., 2018**). Compared to the polluted aerosols with high nitrate level, the

carcinogenic aerosols with great nitrite concentration may be more harmful to human health. As an intermediate in

the photochemistry activities, nitrite appears to be the main product under weak sunlight. The light-dependent




negative correlation between nitrate and nitrite highlights illumination an inducing factor in the atmospheric
nitrogen cycling.
Actually, we discussed the γ-values based on the average experimental results while gave little care to various
measurement errors. Compared to the arithmetic mean results, the percentile γ-values estimated by Monte Carlo
simulation could be more suitable for modelling studies due to the differences between real atmosphere and the
simulated laboratory condition. Furthermore, sensitivity analysis is helpful in explaining the determining factors
involved in the assessment of uptake capacity. Generally, statistical simulation brings about more accurate
evaluation and provides opportunities to explain the model discrepancy for secondary aerosols.

***Data availability.*** All data are available upon request from the corresponding authors.
***Supporting information.*** Sections on particle characterization, experimental setup, pretreatment for in-suit
DRIFTS test, uptake coefficient estimation, field observations, product observations, detailed reactions in
photocatalytic process, photoinduced nitrite oxidation, Raman detection, Sensitivity analysis.
***Author contributions.*** TW designed the experiments and wrote the paper. YYL and YD contributed to the
DRIFTS spectra analysis. YYL and XZF support the field observation. HYC, YQF, MT, and XD assisted the
Raman measurements. YY, KJL, SA, AB, and IN performed the PEC tests. LWZ guided the data analysis and
paper writing. HBF and JMC provided some experimental facilities. All authors were involved in the discussion.
***Competing interests.*** The authors declare no competing financial interests.
***Acknowledgements.*** The authors gratefully acknowledge financial support from Ministry of Science and
Technology of the People's Republic of China (2016YFE0112200, 2016YFC0202700), Marie Skłodowska-Curie
Actions (690958-MARSU-RISE-2015), and National Natural Science Foundation of China (21507011,

353 21677037).

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
