# Peer review of "Is the photochemistry activity weak during haze events?"

_Atmospheric Chemistry and Physics, 2019_

## Referee Comment (RC1) · Anonymous Referee #1 · 13 Jun 2019

This study evaluates the formation of nitrite and nitrate by light-induced heterogeneous reactions of gaseous NO2 on TiO2 used as a proxy for mineral dust. The obtained laboratory results are compared with the behavior of nitrite and nitrate from sampled ambient aerosols. There are number of previous studies on this topic and the results from this study add to the current knowledge on this topic. This study is within the scope of Atmospheric Chemistry and Physics. The work presented here is overall well done, however I am still reluctant to follow the authors' general conclusion in the importance of illumination conditions for nitrite and nitrate formation, mainly due to a

few shortcomings of the study that need to be addressed and clarified before the paper should be considered for publication.

General comments: (1) The light intensities are given in mW cm-2. How were these values measured? Are they integrated values in certain wavelength range? How do they correlate with the solar actinic flux? (2) The NO2 mixing ratio of 15 ppm is extremely high. The authors mentioned Langmuir-Hinshelwood mechanism but in a number of previous studies it was demonstrated that the NO2 uptake coefficients decrease with the increasing NO2 mixing ratios up to 100 ppb. Higher mixing ratios than 100 ppb do not influence the uptake coefficients. (3) Generally, the data presented in this work had no error bars, therefore no uncertainties of those data can be evaluated. I generally feel that more solid evidences are need to arrive the conclusion.

Specific comments:

Line77: It is not mentioned what brand was used for the ultrapure water and how was tested the purity of the water? Line 79: It is not mentioned the purity of the air bottle. What is the level of VOCs in this bottle? Line 119: How was derived this equation? Where does it come from? Line 125: It would be better the equation to be presented as $(2.1 \pm 0.2) 1015$ Line 131: "Based on earlier finds"….should state "Based on earlier findings" Line 169: The wavelength should be $\leq 390$ nm. Line 172: The dot on hydroxyl radical should be on O atom and not on H atom. Line 197-198: These two questions are not properly formulated. For example "Whether" does not fit here. I am not native English speaker but I think the English language usage must be substantially improved in the core of the manuscript. Line 235: On which basis is this statement that "the irradiation tends to be weaker in winter"? Where is this applied? What solar zenith angle, latitude? Etc. Lines 293-295: The authors stated "low and high NO2 concentration of 9.20 ppm and 21.45 ppm. These values are not concentrations but mixing ratios and both values are extremely high. Lines 313-314: This definition is very strange, "polluted aerosols" and "carcinogenic aerosols". Please change this.

Supporting Information

On Y axis should be the intensity or spectral irradiance. It depends what values the authors measured and how were these values measured (see my general comment above).

Table S1: It is stated the velocity of SO2 instead of NO2. Table S2: The presentation of data is not scientific here. Please change.

Conclusions and Implications

I would be very careful to claim the importance of sunlight during the haze events, as you merely tested a few aerosol samples, and they cannot represent all aerosols types. Note your experimental conditions are often not atmospherically relevant (high NO2 mixing ratios and light intensities). The findings should not be over-interpreted and stated with caveats.

---

## Referee Comment (RC2) · Anonymous Referee #2 · 18 Jun 2019

Overview. The paper by Wang et al. mostly describes laboratory experiments and some field data exploring the role of heterogeneous chemistry involving nitrogen dioxide (NO2) on surrogate mineral dust and ambient aerosols. The study is comprised of three parts: (1) an FTIR study of NO2 adsorption to titanium dioxide (TiO2; industrial-grade photocatalyst) in the dark and when irradiated with UV-visible light; (2) development of a parameterization of the NO2 uptake coefficients as a function of light intensity as a potential approach for describing NO2-to-nitrite and nitrate conversion on mineral dust. This work is based on the abovementioned FTIR studies and Monte Carlo simu-

lations were used to estimate error (details on this part are scarce); and (3), results of a field campaign showing that nitrate and nitrite concentration in aerosols are positively correlated at night, but inversely related when they are collected during the day. The authors conclude that FTIR study shows that the nitrite:nitrate ratios observed in ambient aerosol are due to heterogeneous chemistry involving NO2 conversion on mineral dust surfaces containing TiO2. There are several major issues with this manuscript that make it unacceptable for publication. These are outlined below.

General Comments on the Study.

The authors choose as a surrogate for mineral dust Degussa TiO2 which is mixture of rutile and anatase that is designed to be highly reactive. However, the photoactive mineral anatase is one of the scarcest Ti minerals in Earth's crust. For this reason, I feel that Degussa TiO2 is not a suitable surrogate for photochemical studies of mineral dust heterogeneous chemistry. That being said, it is probably as good as SiO2 or alumina for use as a surrogate for non-photochemically active mineral surfaces.

The FTIR product study of NO2 adsorption and photochemistry on TiO2 surfaces is parameterized and used to explain the aerosol field results. However, this presumes that the aerosols collected contain mineral dust containing an appreciable amount of anatase surface sites. Unfortunately, the ambient aerosols have not been characterized (e.g., with elemental analysis and crystallographic methods) and therefore, there is nothing supports the validity of using Degussa anatase to represent the aerosol chemistry occurring in the collected ambient aerosol samples. Indeed, the relationships between nitrate and nitrite for these field samples could as easily be explained through, by now, well established non-TiO2 chemistry. For example, nighttime data can be explained by uptake of NO2 into aqueous droplets, or onto non-TiO2 mineral surfaces, while daytime chemistry can be explained by aerosol phase nitrate photolysis. For example, Xianliang Zhou et al. have published results showing that aerosol nitrate is a major source of nitrite and HONO in aerosols and all evidence suggests that TiO2 is not necessarily needed for this chemistry.

Concentrations of NO2 used for the FTIR study are in the range of 9-21 parts-per-million (ppm). These concentrations are unheard of in the natural or even urban environment unless one considers the chemistry occurring within the engine of a car or a power plant stack. Therefore, such data cannot be extrapolated to environmental conditions where ambient NOx levels are orders of magnitude lower; for this reason, the parameterizations they develop are only applicable to Degussa TiO2 under the conditions of their study. By now it has been well established that at high concentrations of NO2, the heterogeneous mechanism involves dimerization of NO2 followed by autoin-onization of N2O4 and reaction with water (see the review article by Finlayson-Pitts et al, PCCP, 2003, which the authors seem to not be aware of based on their citations). While this mechanism explains much of the thermal chemistry in the manuscript by Wang et al., under ambient concentrations, it is to slow to explain NOx-to-nitrite chemistry under ambient levels of NO2; under ambient conditions, the mechanism likely rather proceeds with the mediation by aerosol components that are more abundant than TiO2 and can also occur in the absence of mineral dust.

In my opinion, the suggested mechanisms are not thought out carefully and in some cases are inaccurate. For example, the authors suggest that negative OH stretches in the FTIR difference spectrum indicate that NO2 conversion to nitrite and nitrate on TiO2 in the dark involves reactivity at the Ti-OH sites (that is involvement of hydride anions), which would imply breaking the Ti-O bond. The energetics of this is likely not favorable based on bond strength considerations; the involvement of water is more likely. Caution should be used in interpreting negative peaks in the OH stretching region in the FTIR difference spectrum as they also develop when the H-bonding environment changes. Again, the authors should refer to the Finlayson-Pitts et al. review article in PCCP (2003) for more discussion for mechanism at such high NO2 concentrations.

The work is also not novel. The TiO2-NOx system has been exhaustively studied by numerous groups over the years in both the catalysis and atmospheric chemistry community. Previous papers presented detailed mechanisms that are not accurately

considered or interpreted in the current manuscript. One interesting discussion is that of how adsorbed nitrite can be oxidized to nitrate by the reaction of NOx on the surface, an observation that is nonetheless well documented in the literature (see numerous papers by Szyani et al.) and is shown to occur under high NOx concentrations by researchers studying catalytic converter technology.

Conclusions are made that are not supported by the results and overstate the importance of the results. For example, starting on line 297 the authors state, "Generally, higher NO2 concentrations suggest broader influence scope of illumination. Hence, current serious NO2 pollution may increase the participation of solar irradiation in the formation of secondary aerosols." It is not clear how this work actually addresses formation of secondary aerosol since most of the work looked at NO2 chemistry on TiO2 and a rather limited study of nitrite and nitrate levels in ambient aerosols. In another example, on line 322, the authors state, "However, the nitrate formation on mineral dust is found to be more dependent on weak sunlight, indicated that photochemistry processes are still crucial in heavy haze." They further state as one of the main conclusions of the paper is that nitrate formation on mineral dust is "more dependent on weak sunlight." Unfortunately, it was unclear to me how the authors can extract this conclusion from their data. It is well known that nitrate formation is driven by photochemical oxidation that converts NO2 to nitric acid/nitrate in a radical termination step. This photochemical pathway is directly proportional to light intensity, which drives both OH and NOx production rates. The nitrate formation rate on TiO2 is also shown to increase with light intensity as shown in Figure 1G.

The authors also suggest there is a "nitrite burst accompanied by low nitrate concentrations" that has not been considered by the atmospheric community. It is not clear what burst they are referring to, as this part of the discussion was quite unclear. The conclusion appears to comes from the correlations observed in Figure 3, but those observations are expected and can be interpreted using known chemistry. For example the positive correlation between nitrite and nitrate at evening/night is expected since

both stem from adsorption of N2O5 and NO2 to particles, while a negative correlation is expected during the day since high photon flux generates high concentrations of more photochemically stable nitric acid/nitrate, but effectively photolyzes nitrite.

In summary, The paper is poorly organized and written; there is no experimental evidence that the ambient aerosol N chemistry is driven by or can be linked to TiO2 (anatase) chemistry; the TiO2 study is not conducted under atmospherically relevant conditions on a substrate that is not atmospherically relevant; and some of the main mechanisms proposed are either inaccurate or lack experimental support and can rather be explained using well-established chemistry. For this reason, I do am unable to recommend this manuscript for further consideration. Moving forward, I would suggest that the authors place all of the TiO2 data (which appears to be of good quality) into a concise manuscript (break the multi-panel figures up into more digestible figures) focused on the topic of photocatalytic reactions of N on TiO2 surfaces, which could be submitted to a more specialized journal in the area of catalysis or environmental remediation.

Specific Comments: Too many figures are shown in Figure 1. These figures can easily be broken up in a way to make a separate manuscript on its own (see above comment).

line 62: The authors state that some researchers have studied the effect of radiation on NO2 uptake on TiO2 but have ignored the reaction mechanism behind the trends in reactivity. However, after reading the papers cited and some that are not cited, I disagree.

line 229-230. In figure 2c, the authors fit a polynomial function to the experimental data collected for NO2 uptake coefficients plotted as a function of light intensity. They then provide an inset showing a near perfect correlation between experimental uptake coefficients and those making up the line they fit to it. This is unnecessary. I recommend omitting the inset and simply reporting the R-squared value for the polynomial fit.

237-238: The polynomial fits lack any physical meaning and are only applicable to the

substrate and conditions of the experiment, which are not necessarily environmentally applicable (see above).

Figure 3: It is entirely unclear what the rightmost column is depicting. What do the authors mean by "nitrite classification?" What do the color codes mean?

---

## Referee Comment (RC3) · Anonymous Referee #3 · 21 Jun 2019

This is a new contribution aiming at understanding the effect of mild illumination on mineral dust proxies and linking those observations with a limited set of environmental data. The claim having motivated this investigation can be found in the fact that previous studies have not seriously discussed the photochemistry under weak sunlight during haze conditions, and have thus ignored some underlying processes. I would certainly agree on such a statement, that I found quite interesting.

However, I'm not fully convinced by the current version of the manuscript that would need significant revision to convey a better and stronger elaborated message.

First of all, I do need to admit that I found it difficult to read this manuscript which contains a series of awkward sentences, such: "The growth of nitrate formation rate tends to be slow after the initial fast with increasing light intensity. For example, the geometric uptake coefficient ($\gamma$geo) under 30.5 mW/cm2 ($5.72\times10$-6) has exceeded the 50 % of that under 160 mW/cm2 ($1.13\times10$-5). This case can be explained by the excess NO2 adsorption under weak illumination while the excess photoinduced active species under strong irradiation. Being negatively associated with nitrate (R2=0.748, P<0.01), nitrite acts as the intermediate and decreases with. . .". This clearly weakens the content of this manuscript.

Then the link being made with the ambient measurements and the test on TiO2 particles is far from being obvious, and would certainly need to more elaborated to make a stronger case.

Most of the data presented have been made at quite high NO2 concentrations (tens of ppm), corresponding to a concentration regime where N2O4 is known to be a significant intermediate for NOx conversion on surfaces. While such concentrations are inherent to the DRIFT technique, one can still wonder how this could affect the findings reported here. It is stated that "Since no saturation effects were observed in the DRIFTS experiments, the NO2 adsorption rate can be regarded as constant.", but this is strong contrast with previous studies dedicated to NO2 heterogeneous chemistry. Could that be to the involvement of N2O4?

In TiO2 driven photocatalysis, it is known that the conversion rate is linear function of light intensity at low intensity, and then levels off at high power to reach steady state conditions. While this seems to be also observed here (inset of figure 2a) but not really discussed nor mentioned, why? In addition, one could argue that the light intensities used here are far from being small and I would not consider this as being photochemistry under mild conditions.

One of the key finding here is associated with the nitrite levels between sustained (or

more important) under mild illumination, but the section "mechanism" does not provide a real explanation for that.

You should try to define what you mean with cumulative uptake coefficients.

A few chemical reactions are described in a simplified i.e., wrong way with unbalanced stoichiometry. This should be avoided.

I did found Figure 3 quite difficult to understand and I am finally unsure about the message the authors wants to convey with this illustration. This maybe needs to be better discussed.

---

## Author Comment (AC1) · 3 Jul 2019

The comment was uploaded in the form of a supplement:
https://www.atmos-chem-phys-discuss.net/acp-2019-315/acp-2019-315-AC1-supplement.zip

---

## Author Comment (AC2) · 3 Jul 2019

The comment was uploaded in the form of a supplement:
https://www.atmos-chem-phys-discuss.net/acp-2019-315/acp-2019-315-AC2-supplement.zip

---

## Author Comment (AC3) · 3 Jul 2019

The comment was uploaded in the form of a supplement:
https://www.atmos-chem-phys-discuss.net/acp-2019-315/acp-2019-315-AC3-supplement.zip

---

## Author Comment (AC4) · 2 Aug 2019

The comment was uploaded in the form of a supplement:
https://www.atmos-chem-phys-discuss.net/acp-2019-315/acp-2019-315-AC4-supplement.pdf

---

## Author Comment (AC5) · 2 Aug 2019

Dear reviewers,

Please find the enclosed response letter for the manuscript titled '*Is the photochemistry activity weak during haze events? —— A novel exploration on the photoinduced heterogeneous reaction of NO₂ on mineral dust*'. The point-by-point replies to the comments are attached on the following pages. We colored **red** in the revised manuscript to the corrections. Thanks a lot for your valuable comments, which greatly improved this work.

Sincerely yours,

Liwu Zhang

Shanghai Key Laboratory of Atmospheric Particle Pollution and Prevention, Department of Environmental Science and Engineering

Fudan University

Shanghai, 200433, P. R. China

E-mail: zhanglw@fudan.edu.cn

**Anonymous Referee #1**

This study evaluates the formation of nitrite and nitrate by light-induced heterogeneous reactions of gaseous $NO_2$ on $TiO_2$ used as a proxy for mineral dust. The obtained laboratory results are compared with the behavior of nitrite and nitrate from sampled ambient aerosols. There are number of previous studies on this topic and the results from this study add to the current knowledge on this topic. This study is within the scope of Atmospheric Chemistry and Physics. The work presented here is overall well done, however I am still reluctant to follow the authors' general conclusion in the importance of illumination conditions for nitrite and nitrate formation, mainly due to a few shortcomings of the study that need to be addressed and clarified before the paper should be considered for publication.

General comments:

(1) The light intensities are given in mW cm$^{-2}$. How were these values measured? Are they integrated values in certain wavelength range? How do they correlate with the solar actinic flux?

Answer:

Thanks a lot for your valuable comments, which greatly improved this work.

The light intensity in laboratory experiments was determined using an optical power meter. We have added details into the method section **(line 106-108)**. *'Herein, ten light intensity levels (0.0, 0.3, 5.4, 17.5, 23.8, 30.5, 54.5, 98.5, 128.1, and 160.0 mW·cm$^{-2}$) were referred based on the measurement by an optical power meter (CEL-NP2000, Beijing Ceaulight Co., LTD, China).'*

The wavelength distribution of the Xenon lamp irradiation is consistent with that of solar irradiation (Figure S3). That is, the light is comparable with the real sunlight. We have explained it in the manuscript **(line 96-98)**. *'A xenon lamp (CEL-TCX250, Beijing Ceaulight Co., LTD, China) was used to provide simulated solar irradiation upon the particles **(Figure S3)**.'*

Additionally, the unit of light intensity is mW/cm$^2$, defined as the solar energy per contact area.

(2) The $NO_2$ mixing ratio of 15 ppm is extremely high. The authors mentioned Langmuir-Hinshelwood mechanism but in a number of previous studies it was demonstrated that the $NO_2$ uptake coefficients decrease with the increasing $NO_2$ mixing ratios up to 100 ppb. Higher mixing ratios than 100 ppb do not influence the uptake coefficients.

Answer: In the revised manuscript, authentic dust (Kaolinite) has been considered and was exposed to a much lower $NO_2$ concentration ($2.21\times10^{13}$ molecules cm$^{-3}$) for better understanding the atmospheric reactions (see section 2.3 and 3.4 in the revised manuscript).

Langmuir-Hinshelwood was mentioned when explaining the nitrite oxidation under dark condition **(line 204-206)**. '*The nitrite would react with another surface nitrite in a Langmuir-Hinshelwood mechanism* **(R.3)** *or gaseous $NO_2$ in an Eley-Rideal mechanism* **(R.4)** *to form nitrate in the absence of illumination* **(Tang et al., 2018; Underwood et al., 1999; Wu et al., 2013)**.'

Among atmospheric laboratory studies (especially DRIFTS experiments), mixing ratios of $NO_2$ at ppm levels are frequently used. The $NO_2$ concentration for $TiO_2$ ($3.77\times10^{14}$ molecules cm$^{-3}$) is much lower than those in recent reports (**Table R1**). We have replaced the unit 'ppm' by 'molecules cm$^{-3}$' when discussing the $NO_2$ concentration.

Additionally, according to the concentration dependence experiments, the photoinduced heterogeneous uptake of $NO_2$ on particles can be viewed as pseudo-first-order reaction (Section S5). Many previous studies confirmed this as well and we have cited these literatures **(Goodman et al., 1998; Guan et al., 2014; Li et al., 2010)**. Hence, the uptake coefficients obtained under high $NO_2$ concentrations are still authentic and valuable for other cases. We have emphasized this in the revised manuscript **(line 287-290)**. '*Since the reaction is first order with respect to $NO_2$ concentration ($2.26\times10^{14}\sim5.27\times10^{14}$ molecules cm$^{-3}$) under various light intensities* **(Figure S8)**, *the γ-values would still be authentic for reactions with lower or higher concentrations* **(Goodman et al., 1998; Guan et al., 2014)**.' In addition, authentic dust (Kaolinite) has been considered and was exposed to a much lower $NO_2$ concentration ($2.21\times10^{13}$ molecules cm$^{-3}$) for better understanding the atmospheric reactions (see section 2.3 and 3.4 in the revised manuscript).

We have read the literatures concerning reactions under atmospheric conditions **(El Zein and Bedjanian, 2012; Ndour et al., 2008; Wang et al., 2012)**. It was reported that the γ-values for the heterogeneous reaction of $NO_2$ on primary aerosols are strongly dependent on $NO_2$ mixing ratios, in the range of several hundred ppb, which can be explained via a Langumuir-Hinshelwood mechanism. We have emphasized this point in the revised manuscript **(line 290-292)**. '*Noticeably, the γ-values decrease with increasing $NO_2$ concentration at atmospheric levels, indicating that the applied γ-values for models may be much greater than the estimated* **(El Zein and Bedjanian, 2012; Ndour et al., 2008; Wang et al., 2012)**.' In addition, reactions with much lower $NO_2$ mixing ratios

(<500 ppb) are extremely difficult to performed in the *in-situ* DRIFTS system (see **Table R1**). Similar studies with low $NO_2$ concentrations were mainly carried out by means of flow tube reactor. We will explore more processes under low concentrations of trace gases in the future.

**Table R1. Summary of similar research on the heterogeneous reaction of $NO_2$ on primary aerosols.**

| Particle type | Main setup | $NO_2$ concentration (molecules $cm^{-3}$) | References |
|---|---|---|---|
| $TiO_2$ | *In-situ* DRIFTS | $3.77 \times 10^{14}$ | **This study** |
| Kaolinite | *Ex-situ* flow reactor | $2.21 \times 10^{13}$ | **This study** |
| $\gamma$-$Al_2O_3$ | *Ex-situ* reactor | $2.46 \times 10^{15}$ | **(Tang et al., 2018)** |
| $\alpha$-$Fe_2O_3$ | *In-situ* DRIFTS | $9.83 \times 10^{14}$ | **(Niu et al., 2017)** |
| $CaCO_3$ | FTIR, Raman | $2.60 \times 10^{15}$ | **(Tan et al., 2017)** |
| $CaCO_3$ | *In-situ* DRIFTS | $2.60 \times 10^{15}$ | **(Tan et al., 2016)** |
| $\gamma$-$Al_2O_3$ | *In-situ* DRIFTS | $1.12 \times 10^{15}$ | **(Sun et al., 2016)** |
| $\alpha$-$Fe_2O_3$ | Flow tube reactor | $3.81 \times 10^{12}$ | **(Liu et al., 2015)** |
| $\gamma$-$Al_2O_3$ | *In-situ* DRIFTS | $1.12 \times 10^{15}$ | **(Sun et al., 2015)** |
| NaCl | *In-situ* DRIFTS | $2.7 \times 10^{15}$ | **(Chen and Zhu, 2014)** |
| $\alpha$-$Al_2O_3$ | *In-situ* DRIFTS | $2.69 \times 10^{14}$-$5.38 \times 10^{15}$ | **(Guan et al., 2014)** |
| $\alpha$-$Al_2O_3$ | *In-situ* DRIFTS | $1.21 \times 10^{15}$ | **(Wu et al., 2013)** |

(3) Generally, the data presented in this work had no error bars, therefore no uncertainties of those data can be evaluated. I generally feel that more solid evidences are need to arrive the conclusion.

Answer:

In this study, each independent variable was determined via five or more replication measurements, followed by the Monte Carlo (MC) simulation to present the data in a more accurate way. Based on these, percentile values of the γ-values can be obtained and have been presented in Figure 4 and 5 of the revised manuscript. However, for better comparisons with experimental results in other studies, arithmetic mean values were used as well. We have emphasized this point in section

2.6 of the revised manuscript **(line 154-155)**. '*Arithmetic mean values are used in the following comparisons for better understanding*'. Hence, the averaged γ-values were shown in the current Figure 4 and 5 along with the cumulative probability distribution of the uptake coefficients. We emphasized this point in the captions of Figure 4 and 5 **(line 319 and 354-355)**. '*Insets reveal the actual γ-values (Mean±1σ) and theoretical ones*'; '*Actual γ-values (Mean±1σ) coupled with estimated dγ$_{geo}$/dI*'.

Unfortunately, the arithmetic mean values were not put into Table S3. To facilitate comparison, we added them into Table S3 and illustrated this in the revised manuscript **(Line 285-287)** '*To accurately evaluate the photoinduced nitrate formation, γ-values were estimated based on Monte Carlo simulation with the cumulative probability distributions depicted in* **Figure 4**. *Percentile values and arithmetic mean values are summarized in* **Table S3**.' The values for Kaolinite particles have been presented in **Table S4**.

**In the revised manuscript, some more evidences have been provided to support the conclusions.**

**Firstly**, $Al_2O_3$ has been employed as the surrogate of atmospheric non-photoactive component. The association between nitrite and nitrate was positive on $Al_2O_3$, while appears to be negative on $TiO_2$, resulting mainly from the photolysis of nitrogen compounds and the photoinduced oxidation of nitrite intermediates, respectively. DRIFTS spectra on the irradiated '$TiO_2$-$NaNO_2$' and '$Al_2O_3$-$NaNO_2$' could emphasize these results. More details have been presented in Section 3.2 of the revised manuscript.

**Secondly**, authentic dust (Kaolinite) has been considered to make the conclusions more believable. We have introduced it in the revised manuscript **(line 323-326)**. '*Although being ubiquitous in natural environment, $TiO_2$ is in representative of single-component mineral dust. Kaolinite, comprising $SiO_2$, $Al_2O_3$, $Fe_2O_3$, $TiO_2$ and other components (Table S1), is a more typical authentic mineral dust.*' In section 3.4 of the revised manuscript, the uneven promotion effect can be observed on Kaolinite as well, along with the negative association between nitrite and nitrate products. The negative association is similar with that on $TiO_2$ while differs from that on $Al_2O_3$, indicating that the atmospheric photocatalytic components are strongly associated with nitrogen chemistry. Furthermore, a much lower $NO_2$ concentration ($2.21×10^{13}$ molecules cm$^{-3}$) has been used

to make the experimental situation closer to the atmospheric environment. More details are presented in section 2.3 and 3.4 of the revised manuscript.

**Thirdly**, more probable reaction mechanisms have been considered and added into the revised manuscript. The photolysis of nitrogen compounds has been discussed in section 3.1. The probable processes linked to $N_2O_4$ has been introduced in section 3.2. The photolysis processes have been introduced when explaining the field observation results.

**Finally**, we have also consulted native speaker to improve the language. The novelty of the research, as well as some other valuable points have been highlighted in the revised manuscript. We believe this article an important progress in the atmospheric heterogeneous research.

Specific comments:

Line77: It is not mentioned what brand was used for the ultrapure water and how was tested the purity of the water?

Answer:

We have added the description for ultrapure water in the revised manuscript **(line 87-88)**. '*Ultrapure water (specific resistance ≧18.2 MΩ cm) produced from a deionizer (Direct-Q5-UV, MERCK, Germany) was used throughout the research process.*'

A conductivity meter was coupled with the water purifier to ensure the specific resistance above *18.2 MΩ cm.*

Line 79: It is not mentioned the purity of the air bottle. What is the level of VOCs in this bottle?

Answer:

The purity of the $N_2$ and $O_2$ is 99.9999%, and the purity of $NO_2$ calibration gas is 99.999%. We have added the purity information in the revised manuscript **(line 89-91)**. *'High-pure air (79% $N_2$ and 21% $O_2$, 99.9999% purity, Shanghai TOMOE Co., LTD, China) and $7.37×10^{15}$ molecules $cm^{-3}$ $NO_2$ ($N_2$ dilution, 99.999% purity, Shanghai Qingkuan Co., LTD, China) were included in this research.'*

Scarcely any impurities (e.g. VOCs) were included because the high-pure air was sent into molecular sieve for purification before coming into the gas supply system **(line 91-92)**. *'Prior to*

*coming into the gas supply system, high-pure air went through silica gel and molecular sieve for*

*drying and purification, respectively.'*

Line 119: How was derived this equation? Where does it come from?

Answer:

The reactive uptake coefficients (γ) for the heterogeneous adsorption of tracer gases were deduced from the DRIFTS experiments, and is frequently used in the similar studies **(Li et al., 2006; Wu et al., 2011)**. We have introduced the definition in the revised manuscript **(line 133-134)**. *'The reactive uptake coefficient, γ, is defined as the ratio of the reactive gas-surface collision rate $(d[NO_3^-]/dt)$ to the total gas-surface collision rate (Z)* **(Gustafsson et al., 2006; Tan et al., 2017; Tan et al., 2016)**. *'*

Line 125: It would be better the equation to be presented as (2.1 ± 0.2) 1015

Answer:

We have changed the sentence to '*The conversion factor (f), estimated to be $(2.09 \pm 0.16) \times 10^{15}$ (ion · K-M unit$^{-1}$) in this study.* ' **(line 145)**

Line 131: "Based on earlier finds": : :.should state "Based on earlier findings"

Answer:

'Based on earlier findings' is the correct expression. However, we have added more details into the section **(line 153-154)**. The original sentence has been changed into *'We performed independent runs at 1500, 3000, 5000, and 10000 iterations with each parameter. The results showed that 5000 iterations are sufficient to ensure the stability of the results.'*

Line 169: The wavelength should be ≤ 390 nm.

Answer:

We have changed it to '*wavelength≤390 nm*' in the revised manuscript **(line 193)**.

Line 172: The dot on hydroxyl radical should be on O atom and not on H atom.

Answer:

We have changed it into ''*OH*' **(line 196)**.

Furthermore, some similar errors have been corrected in the revised Supporting Information (Section S8).

Line 197-198: These two questions are not properly formulated. For example "Whether" does not fit here. I am not native English speaker but I think the English language usage must be substantially improved in the core of the manuscript.

Answer:

We have revised them. Actually, we added another section discussing the reactions on authentic dust Kaolinite. These two questions were inferred in the end of the section 3.4 of the revised manuscript **(line 348-351)**. '*Generally, two questions become more interesting when discussing the heterogeneous process on mineral surrogates or authentic dust: What is the association between nitrate and nitrite in atmospheric particulates? What is the proposed mechanism behind these illumination effects? We may discuss these issues in the following sections.*'

We have also consulted native speaker to improve the language.

Line 235: On which basis is this statement that "the irradiation tends to be weaker in winter"? Where is this applied? What solar zenith angle, latitude? Etc.

Answer:

According to the regression analysis on the uptake coefficients, the $d\gamma_{geo}/dI$ values can be obtained by derivation to distinguish the illumination effects varying with light intensity. The solar irradiation intensities of Shanghai, China was adopted as an example for better comparisons as explained in the revised manuscript **(line 311-313)**. '*In Shanghai (121°29'E, 31°11'N), the 3h-average solar irradiation intensities are mostly lower than 80 mW/cm$^2$ (NOAA data, https://www.arl.noaa.gov/), indicating noticeable sunlight impacts.*'

The comparisons between solar intensities in cold and warm seasons can be obtained from Figure 4d. We have revised the legend of Figure 4d for better understanding.

Lines 293-295: The authors stated "low and high $NO_2$ concentration of 9.20 ppm and 21.45 ppm. These values are not concentrations but mixing ratios and both values are extremely high.

We have replaced the unit 'ppm' by 'molecules cm$^{-3}$' when discussing the NO$_2$ concentration.

Herein, the NO$_2$ concentrations are higher than the common levels in the troposphere. However, as summarized by **Table R1**, NO$_2$ concentrations in laboratory studies (especially *in-situ* DRIFTS experiments) are mostly higher than those in this work. To make the experimental situation closer to the real atmosphere, we conducted experiments on authentic dust (Kaolinite) under a much lower NO$_2$ concentration ($2.21 \times 10^{13}$ molecules cm$^{-3}$). More details can be found in the added Section 2.3 and 3.4 in the revised manuscript.

Lines 313-314: This definition is very strange, "polluted aerosols" and "carcinogenic aerosols". Please change this.

Answer:

We have changed the definitions in the revised manuscript **(line 439-440)**. 'Nitrite-rich and nitrate-rich products may occupy the mainstream under faint sunlight and strong irradiation, respectively.'

Supporting Information

On Y axis should be the intensity or spectral irradiance. It depends what values the authors measured and how were these values measured (see my general comment above).

Answer:

Figure S3 presents the spectral distribution of the Xenon lamp light measured by a fiber optic spectrometer (AULTT-P4000, Beijing Ceaulight Co., LTD, China).

For the Y axis, the original 'Intensity' should be changed to 'Relative intensity' to represent the relative spectral distribution under any light intensity.

Table S1: It is stated the velocity of SO$_2$ instead of NO$_2$.

Answer:

It is a mistake. We have changed 'SO$_2$' to 'NO$_2$'.

Table S2: The presentation of data is not scientific here. Please change.

Answer:

We have added the Mean±SD into Table S2.

Conclusions and Implications

I would be very careful to claim the importance of sunlight during the haze events, as you merely tested a few aerosol samples, and they cannot represent all aerosols types. Note your experimental conditions are often not atmospherically relevant (high $NO_2$ mixing ratios and light intensities). The findings should not be over-interpreted and stated with caveats.

Answer:

Firstly, we did statistical tests on the field data, with results ($p$ value and $R^2$) present in Figure 4 of the revised manuscript. We will do more field observations in the future to support the results.

Secondly, the $NO_2$ concentrations in the original manuscript is higher than atmospheric levels while lower than most selections in laboratory research (see Table R1). In the revised manuscript, we have added many experimental results to illustrate the heterogeneous process on authentic dust (Kaolinite) with a much lower $NO_2$ concentration ($2.21 \times 10^{13}$ molecules $cm^{-3}$). We think the '*authentic dust-low $NO_2$ concentration*' system could provide strong evidences to the uneven promotion effect on nitrate formation and the negative dependence between nitrite and nitrate in the atmosphere. The experimental results on Kaolinite are shown in Section 2.3 and 3.4 of the revised manuscript.

Moreover, light intensities higher than 100 mW/cm$^2$ were adopted for $TiO_2$ particles to explore the main reaction mechanism. For the authentic dust Kaolinite, the light intensities for all tests were lower than 100 mW/cm$^2$ **(line 113-114)**. '*In each test, 50 mg Kaolinite was put into the chamber, followed by the exposure to $2.21 \times 10^{13}$ molecules $cm^{-3}$ $NO_2$ under diverse light intensities (0, 10, 40, 70, and 100 mW/cm$^2$).*'

Based on these findings, we can emphasize the importance of light intensity when considering the photochemistry process in atmosphere, which has been confirmed in the current study on the uneven illumination effects in the formation of nitrate and nitrite.

**Anonymous Referee #2**

Overview. The paper by Wang et al. mostly describes laboratory experiments and some field data exploring the role of heterogeneous chemistry involving nitrogen dioxide ($NO_2$) on surrogate mineral dust and ambient aerosols. The study is comprised of three parts: (1) an FTIR study of $NO_2$ adsorption to titanium dioxide ($TiO_2$; industrial grade photocatalyst) in the dark and when irradiated with UV-visible light; (2) development of a parameterization of the $NO_2$ uptake coefficients as a function of light intensity as a potential approach for describing $NO_2$-to-nitrite and nitrate conversion on mineral dust. This work is based on the abovementioned FTIR studies and Monte Carlo simulations were used to estimate error (details on this part are scarce); and (3), results of a field campaign showing that nitrate and nitrite concentration in aerosols are positively correlated at night, but inversely related when they are collected during the day. The authors conclude that FTIR study shows that the nitrite: nitrate ratios observed in ambient aerosol are due to heterogeneous chemistry involving $NO_2$ conversion on mineral dust surfaces containing $TiO_2$. There are several major issues with this manuscript that make it unacceptable for publication. These are outlined below.

General Comments on the Study.

The authors choose as a surrogate for mineral dust Degussa $TiO_2$ which is mixture of rutile and anatase that is designed to be highly reactive. However, the photoactive mineral anatase is one of the scarcest Ti minerals in Earth's crust. For this reason, I feel that Degussa $TiO_2$ is not a suitable surrogate for photochemical studies of mineral dust heterogeneous chemistry. That being said, it is probably as good as $SiO_2$ or alumina for use as a surrogate for non-photochemically active mineral surfaces.

Answer:

Thanks a lot for your valuable comments, which greatly improved this work.

We notice your concern on employing anatase $TiO_2$ as the surrogate of mineral dust. In the revised manuscript, we have explained the reason and further studied $Al_2O_3$ and authentic dust (Kaolinite) as well to support the conclusions.

**Firstly**, titanium dioxide ($TiO_2$) is ubiquitous in the atmosphere, originating mainly from natural processes and human activities **(Chen et al., 2012)**. The suspended $TiO_2$ is most likely from

windblown mineral dust. Approximately 0.7 wt % $TiO_2$ is found in the continental dust **(WEDEPOHL, 1995)**. In summary, $TiO_2$ is found in mineral dust at mass mixing ratios ranging from 0.1% to 10% depending on the exact location from where the particles were uplifted **(Ndour et al., 2008)**. Besides the $TiO_2$ in mineral dust, there are additional sources of airborne PM containing $TiO_2$ from industrial processes, including the nanotechnology industry, that could contribute significantly to $TiO_2$ in the atmosphere. Hence, $TiO_2$ was employed to represent the ubiquitous atmospheric metal oxides.

**Secondly**, $TiO_2$ plays a vital role in the oxidative processes on mineral dust. Park et al, explored the heterogeneous photo-oxidation of $SO_2$ in the presence of Gobi and Arizona dust, and pointed out that the photocatalysis is mainly dependent on $TiO_2$ and $Fe_2O_3$ components **(Park et al., 2017)**. Ponczek et al., investigated the photooxidation of butanol on mineral dust and considered $TiO_2$ the main effective component in the primary particles **(Ponczek and George, 2018)**. Chen et al., verified the essential role of $TiO_2$ during the photooxidation of *m*-xylene with $NO_x$ **(Chen et al., 2018)**. Additionally, many previous studies focused on pure $TiO_2$ or Degussa $TiO_2$ during the atmospheric research **(El Zein and Bedjanian, 2012; Kebede et al., 2013; Tang et al., 2016)**. Generally, $TiO_2$ is responsible for the photochemical reactivity of atmospheric mineral aerosols.

Despite the greater shares of $SiO_2$ and $Al_2O_3$ in atmospheric mineral dust, there is no generation of electron-hole pairs in the conduction and valence bands of these metal oxides under solar irradiation. Research on $SiO_2$ or $Al_2O_3$ particles may neglect the photochemical processes on mineral dust. This study is aimed at exploring the photoinduced formation of nitrate and nitrite on mineral dust. We have added some information on authentic dust and highlighted the significance of $TiO_2$ in mineral dust into the Introduction section **(line 48-61)**. *'As mentioned above, numerous laboratory studies concerned the surface chemistry on mineral dust. Field samples (e.g. Arizona dust, Illite and Kaolinite) are frequently used in relevant research* **(Fu et al., 2010; Huang et al., 2015; Liu et al., 2015)**. *In authentic dust, the major components (e.g. $SiO_2$ and $Al_2O_3$) were always selected as the surrogates to present the general chemical processes, while some photoactive components in small proportions (e.g. $TiO_2$ and $Fe_2O_3$) have direct environmental implications due to their Photocatalysis activity* **(Nanayakkara et al., 2014; Park et al., 2017)**. *$TiO_2$ is found in mineral dust at mass mixing ratios ranging from 0.1% to 10% depending on the exact location from*

*where the particles were uplifted* (**Ndour et al., 2008**). *Prior studies have indicated that the photoinduced oxidation of trace gases by TiO₂ plays an essential role in the chemical balance of the atmosphere* (**Chen et al., 2012**). *TiO₂-coated surfaces are frequently used in the vicinity of buildings or urban infrastructure* (**Ballari et al., 2010; Ballari et al., 2011**). *It is suggested that these self-cleaning materials have a great chance of entering into the environment and then triggering series of atmospheric reactions* (**Balajka et al., 2018; Langridge et al., 2009; Monge et al., 2010; Yang et al., 2018**). *Accordingly, TiO₂ is usually representative of the ubiquitous photoactive components in atmospheric environment* (**Kebede et al., 2013; Moon et al., 2018; Ponczek and George, 2018**).*'*

**Thirdly**, Al$_2$O$_3$ was adopted as the surrogate of atmospheric non-photoactive components. Influenced by the photolysis of nitrogen compounds, the association between nitrite and nitrate is positive on Al$_2$O$_3$, differing from the negative dependence on TiO$_2$. Hence, the photoinduced oxidation of nitrite intermediates is a meaningful phenomenon on atmospheric particles containing photoactive components. DRIFTS spectra could confirm the process as well. More information can be obtained from the new section 3.2 in the revised manuscript.

**Last but not the least**, in this work TiO$_2$ was firstly used as single-component metal oxide to explore some general chemical rules. Based on the obtained results, authentic dust (Kaolinite) was taken into consideration under a much lower NO$_2$ concentration ($2.21 \times 10^{13}$ molecules cm$^{-3}$) to evaluate the heterogeneous processes in the atmospheric environment. We have highlighted this point in the revised manuscript (**line 324-326**). '*Although being ubiquitous in natural environment, TiO₂ is in representative of single-component mineral dust. Kaolinite, comprising SiO₂, Al₂O₃, Fe₂O₃, TiO₂ and other components (Table S1), is a more typical authentic mineral dust.*' The uneven promotion effect under illumination and the negative correlation between nitrite and nitrate can be found on the authentic dust as well. Noticeably, the association between nitrite and nitrate products on Kaolinite is similar with that on TiO$_2$ while differs from the situation on Al$_2$O$_3$, indicating that the photoinduced oxidation of nitrite intermediates makes great contributions to the nitrate production on atmospheric particles. Hence, TiO$_2$ could also act as a suitable surrogate of atmospheric mineral dust when discussing the atmospheric photoreactions. More details have been included in section 2.3 and 3.4 of the revised manuscript.

The FTIR product study of $NO_2$ adsorption and photochemistry on $TiO_2$ surfaces is parameterized and used to explain the aerosol field results. However, this presumes that the aerosols collected contain mineral dust containing an appreciable amount of anatase surface sites. Unfortunately, the ambient aerosols have not been characterized (e.g., with elemental analysis and crystallographic methods) and therefore, there is nothing supports the validity of using Degussa anatase to represent the aerosol chemistry occurring in the collected ambient aerosol samples. Indeed, the relationships between nitrate and nitrite for these field samples could as easily be explained through, by now, well established non-$TiO_2$ chemistry. For example, nighttime data can be explained by uptake of $NO_2$ into aqueous droplets, or onto non-$TiO_2$ mineral surfaces, while daytime chemistry can be explained by aerosol phase nitrate photolysis. For example, Xianliang Zhou et al. have published results showing that aerosol nitrate is a major source of nitrite and HONO in aerosols and all evidence suggests that $TiO_2$ is not necessarily needed for this chemistry.

Answer: To further support our conclusions, we have added experiments on $Al_2O_3$ and authentic dust (Kaolinite) in the revised manuscript.

In the revised manuscript, $TiO_2$ and $Al_2O_3$ were adopted as the surrogates of atmospheric photoactive and non-photoactive components, respectively. The association between nitrite and nitrate products is positive on $Al_2O_3$ while appears to be negative on $TiO_2$. The positive dependence on $Al_2O_3$ can be explained by the photolysis of nitrogen compounds, whereas the negative one on $TiO_2$ is significantly influenced by the photoinduced oxidation of nitrite intermediates, which can be verified by DRIFTS spectra as well. We have added the results on $Al_2O_3$ in the revised manuscript **(section 3.2)**.

Furthermore, the latest results on authentic dust (Kaolinite) have been added into the revised manuscript **(section 2.3 and 3.4)**. We have introduced this point in the revised manuscript **(line 324-326)**. '*Although being ubiquitous in natural environment, $TiO_2$ is in representative of single-component mineral dust. Kaolinite, comprising $SiO_2$, $Al_2O_3$, $Fe_2O_3$, $TiO_2$ and other components (Table S1), is a more typical authentic mineral dust.*' The uneven promotion effect on nitrate formation can be observed on Kaolinite as well. More importantly, the association between nitrite and nitrate products appears to be negative on Kaolinite, in consistent with that on $TiO_2$, indicating that the photoinduced oxidation of nitrite intermediates makes great contributions to the nitrate

production on authentic dust. Hence, the illumination effects found on $TiO_2$ may exist in the atmospheric environment. Generally, $TiO_2$ could be regarded as the surrogate of mineral dust when discussing atmospheric photoreactions.

For the oxidation of nitrite to nitrate, the promoters are not limited to the free radicals from irradiated $TiO_2$ or $Fe_2O_3$. More atmospheric oxidants may also participate into the process under illumination. Hence, the oxidation of nitrite intermediates can be regarded as a common phenomenon. Based on the knowledge from these laboratory experiments, we tried to explain the field observations. The positive association between nitrite and nitrate during daytime is attributed to the $NO_2$ disproportionation process, which takes place on particle surfaces, in droplets or during the interaction with water vapor, regardless of the presence of $TiO_2$. Generally, we have improved the explanations in the revised manuscript **(line 364-369)**. '*The associations in droplet (Figure 4d-f) and condensation (Figure 4g-i) modes are comparable with those on coarse mode. Inspired by the discussion above, the positive associations during nighttime are attributed to the $NO_2$ disproportionation process on particle surfaces, in suspended droplets, or during the collisions with water vapour. In the presence of solar irradiation, the negative dependence can be explained by the oxidation of nitrite intermediates promoted by various active species in different media.*'

For the negative dependence between nitrite and nitrate aerosols in the presence of sunlight, the photolysis of atmospheric nitrogen compounds seems to be a nice explanation. We have noticed two papers focusing on the photolysis of nitrate and nitrite aerosols **(Ye et al., 2017; Ye et al., 2016)**, as well as some similar studies on this topic **(Benedict et al., 2017; Gen et al., 2019; Kim et al., 2014; Lesko et al., 2015; Nanayakkara et al., 2014; Roca et al., 2008; Rubasinghege and Grassian, 2009; Schuttlefield et al., 2008)**. The photolysis of nitrogen compounds may lead to the negative dependence in another aspect, and could result in the low nitrite levels accompanied by slight nitrate pollution in the presence of sunlight. We have emphasized these valuable points in the revised manuscript **(line 380-386)**. *'It is universally acknowledged that atmospheric nitrate and nitrite may undergo photolysis under acidic conditions. These two processes (photolysis of nitrogen compounds and oxidation of nitrite) may contribute to the negative associations in different aspects. Yet, the nitrite oxidation seems to be more easily to take place because some photoactive components in the atmosphere can be excited by visible light, while the photolysis mainly occurs under ultraviolet light.*

*The exact contributions of these processes to the negative correlation are still not very clear. However, the photolysis of atmospheric nitrate and nitrite is a strong evidence in explaining the low nitrite levels accompanied by slight nitrate pollution in the presence of sunlight.'*

Meanwhile, we have introduced the photolysis processes in section 3.1 of the revised manuscript **(line 210-217)**. *'Under irradiation, nitrite signal gradually fades away and diverse nitrate species make steady contributions to the total products during the photoreactions* ***(Figure S11b)****. Photolysis of nitrogen compounds become a noticeable process after the exposure to sunlight. In summary, photolysis of nitrate produces reactive nitrogen and oxygen species via three different channels* **(Benedict et al., 2017; Gen et al., 2019; Nanayakkara et al., 2014; Rubasinghege and Grassian, 2009; Schuttlefield et al., 2008)**, *forming $NO_2$ (R.5), ˙OH (R.6), and peroxynitrite (R.7). As previously reported, the nitrite products may react with ˙OH to form $NO_2$ (R.8)* **(Kim et al., 2014; Lesko et al., 2015; Roca et al., 2008)**. *HONO release was frequently found in acidic media (R.9), followed by the production of ˙OH (R.10) under irradiation* **(Ye et al., 2017; Ye et al., 2016)**. *'*

Concentrations of $NO_2$ used for the FTIR study are in the range of 9-21 parts-permillion (ppm). These concentrations are unheard of in the natural or even urban environment unless one considers the chemistry occurring within the engine of a car or a power plant stack. Therefore, such data cannot be extrapolated to environmental conditions where ambient $NO_x$ levels are orders of magnitude lower; for this reason, the parameterizations they develop are only applicable to Degussa $TiO_2$ under the conditions of their study.

By now it has been well established that at high concentrations of $NO_2$, the heterogeneous mechanism involves dimerization of $NO_2$ followed by autoinonization of $N_2O_4$ and reaction with water (see the review article by Finlayson-Pitts et al, PCCP, 2003, which the authors seem to not be aware of based on their citations). While this mechanism explains much of the thermal chemistry in the manuscript by Wang et al., under ambient concentrations, it is to slow to explain NOx-to-nitrite chemistry under ambient levels of NO2; under ambient conditions, the mechanism likely rather proceeds with the mediation by aerosol components that are more abundant than TiO2 and can also occur in the absence of mineral dust.

In my opinion, the suggested mechanisms are not thought out carefully and in some cases are

inaccurate. For example, the authors suggest that negative OH stretches in the FTIR difference spectrum indicate that $NO_2$ conversion to nitrite and nitrate on $TiO_2$ in the dark involves reactivity at the Ti-OH sites (that is involvement of hydride anions), which would imply breaking the Ti-O bond. The energetics of this is likely not favorable based on bond strength considerations; the involvement of water is more likely. Caution should be used in interpreting negative peaks in the OH stretching region in the FTIR difference spectrum as they also develop when the H-bonding environment changes. Again, the authors should refer to the Finlayson-Pitts et al. review article in PCCP (2003) for more discussion for mechanism at such high NO2 concentrations.

Answer: In the revised manuscript, authentic dust (Kaolinite) has been considered and was exposed to a much lower $NO_2$ concentration ($2.21 \times 10^{13}$ molecules $cm^{-3}$) for better understanding the atmospheric reactions (see section 2.3 and 3.4 in the revised manuscript).

$NO_2$ with mixing ratios of ppm levels are frequently used among atmospheric laboratory studies. According to **Table R1**, the $NO_2$ concentration for $TiO_2$ is much lower than those in previous studies. In addition, the photoinduced heterogeneous uptake of $NO_2$ on $TiO_2$ particles can be viewed as pseudo-first-order reaction. Hence, the γ-values would still be authentic for reactions with lower or higher $NO_2$ concentrations. Similar results have been confirmed in previous studies and we have cited these literatures to support our findings **(Goodman et al., 1998; Guan et al., 2014)**. More details have been explained in the revised manuscript **(line 287-290)**. '*Since the reaction is first order with respect to $NO_2$ concentration ($2.26 \times 10^{14} \sim 5.27 \times 10^{14}$ molecules $cm^{-3}$) under various light intensities (Figure S8), the γ-values would still be authentic for reactions with lower or higher concentrations* **(Goodman et al., 1998; Guan et al., 2014)**.' In addition, a lower $NO_2$ concentration ($2.21 \times 10^{13}$ molecules $cm^{-3}$) has been employed on authentic particles. The results on Kaolinite were similar with those of the $TiO_2$ particles. In general, the high $NO_2$ concentrations may not influence the kinetics evaluation, and the much lower concentration has been used on authentic dust in order to confirm the illumination effects.

Actually, we have read the literatures concerning reactions under atmospheric conditions **(El Zein and Bedjanian, 2012; Ndour et al., 2008; Wang et al., 2012)**. It has been reported that the γ-values for the heterogeneous reaction of $NO_2$ on primary aerosols are strongly dependent on $NO_2$ mixing ratios, in the range of hundreds of ppb, which can be explained via a Langumuir-Hinshelwood mechanism. We have emphasized this point in the revised manuscript **(line 290-292)**.

*'Noticeably, the γ-values decrease with increasing $NO_2$ concentration at atmospheric levels, indicating that the applied γ-values for models may be much greater than the estimated* **(El Zein and Bedjanian, 2012; Ndour et al., 2008; Wang et al., 2012)**. *'*Reactions with much lower $NO_2$ mixing ratios (<500 ppb) are extremely difficult to carry out in the *in-situ* DRIFTS system (see **Table R1**). Similar studies with extremely low $NO_2$ concentrations were mainly carried out by means of flow tube reactor. We will explore more processes under low concentrations of trace gases in the future.

**Table R1. Summary of similar research on the heterogeneous reaction of $NO_2$ on primary aerosols.**

| Particle type | Main setup | $NO_2$ concentration (molecules $cm^{-3}$) | References |
|---|---|---|---|
| $TiO_2$ | *In-situ* DRIFTS | $3.77 \times 10^{14}$ | This study |
| Kaolinite | *Ex-situ* flow reactor | $2.21 \times 10^{13}$ | This study |
| γ -$Al_2O_3$ | *Ex-situ* reactor | $2.46 \times 10^{15}$ | (Tang et al., 2018) |
| α-$Fe_2O_3$ | *In-situ* DRIFTS | $9.83 \times 10^{14}$ | (Niu et al., 2017) |
| $CaCO_3$ | FTIR, Raman | $2.60 \times 10^{15}$ | (Tan et al., 2017) |
| $CaCO_3$ | *In-situ* DRIFTS | $2.60 \times 10^{15}$ | (Tan et al., 2016) |
| γ -$Al_2O_3$ | *In-situ* DRIFTS | $1.12 \times 10^{15}$ | (Sun et al., 2016) |
| α-$Fe_2O_3$ | Flow tube reactor | $3.81 \times 10^{12}$ | (Liu et al., 2015) |
| γ -$Al_2O_3$ | *In-situ* DRIFTS | $1.12 \times 10^{15}$ | (Sun et al., 2015) |
| NaCl | *In-situ* DRIFTS | $2.7 \times 10^{15}$ | (Chen and Zhu, 2014) |
| α-$Al_2O_3$ | *In-situ* DRIFTS | $2.69 \times 10^{14}$-$5.38 \times 10^{15}$ | (Guan et al., 2014) |
| α-$Al_2O_3$ | *In-situ* DRIFTS | $1.21 \times 10^{15}$ | (Wu et al., 2013) |

$N_2O_4$ (~1700 and 1300 $cm^{-1}$) was not observed in the DRIFTS spectra. We assumed that the $N_2O_4$ existed in the initial stage of the whole reaction and was consumed immediately along with the oxidation of nitrite species. Liu et al., observed the similar phenomenon **(Liu et al., 2012)**. We have added the possible mechanism about $N_2O_4$ in the revised manuscript **(line 224-229)**. *'Different with previous studies, the dimer of $NO_2$, namely $N_2O_4$, was not observed in these spectra around*

*1745 and 1295 cm$^{-1}$. It has been confirmed that N$_2$O$_4$ could isomerize and autoionize to NO$^+$NO$_3^-$, and then react with adsorbed water to generate HNO$_3$ and HONO* **(Syomin and Finlayson-Pitts, 2003)**. *Meanwhile, N$_2$O$_4$ may exist in the initial stage of the whole reaction and be consumed immediately along with the oxidation of nitrite species. Liu et al., investigated the consumption of N$_2$O$_4$ during the oxidation of adsorbed SO$_2$* **(Liu et al., 2012)**. *Herein, the proposed process can be described by (R.6-7).'* Furthermore, Kaolinite particles have been referred to explain the relevant mechanism. Hence, the mechanism would take place on both mineral surrogates and authentic dust.

Actually, TiO$_2$ is fairly stable in chemical processes. We didn't mention any information on the OH stretches.

The work is also not novel. The TiO2-NOx system has been exhaustively studied by numerous groups over the years in both the catalysis and atmospheric chemistry community. Previous papers presented detailed mechanisms that are not accurately considered or interpreted in the current manuscript. One interesting discussion is that of how adsorbed nitrite can be oxidized to nitrate by the reaction of NOx on the surface, an observation that is nonetheless well documented in the literature (see numerous papers by Szyani et al.) and is shown to occur under high NO$_x$ concentrations by researchers studying catalytic converter technology.

Answer:

We have emphasized the innovation and importance of this study in the revised manuscript **(line 66-70)**. '*Most remarkable studies concerned the photocatalytic effects under ultraviolet light while the intensity dependence under simulated solar irradiation was rarely explored* **(Dupart et al., 2014; Guan et al., 2014; Li et al., 2010; Shang et al., 2010)**. *Some researchers* **(El Zein and Bedjanian, 2012)** *noticed the uneven illumination effect on the heterogeneous reaction of NO$_2$ on TiO$_2$ particles while ignored the reaction process on authentic dust and the reaction mechanism behind the variation.*'

TiO$_2$ and Al$_2$O$_3$ were selected as the surrogates of atmospheric photoactive and non-photoactive components, respectively. For the nitrite and nitrate products under diverse light intensities, the association is positive on Al$_2$O$_3$, while appears to be negative on TiO$_2$ particles. The former dependence can be explained by the photolysis of nitrogen compounds, while the negative correlation is attributed to the photoinduced oxidation of nitrite species to nitrate. The oxidation of

nitrite intermediates can be observed on DRIFTS spectra as well. Nitrite oxidation frequently occurs in aqueous media while was scarcely reported on aerosol surfaces **(Hofer, 2018; Sun et al., 2017)**. More details are presented in section 3.2 of the revised manuscript.

In addition, we further employed authentic dust (Kaolinite) and a much lower $NO_2$ concentration ($2.21 \times 10^{13}$ molecules $cm^{-3}$) to explain the uneven promotion effect under illumination and the negative dependence between nitrate and nitrite in the revised manuscript (section 2.3 and 3.4), which presents novelty as well. More importantly, photochemistry process is usually considered weak during heavy haze events, however, it is shown in this work that the photochemistry can also be as strong as that during clear sunny days.

We have read many papers from Szyani's groups **(Desikusumastuti et al., 2008; Ozensoy et al., 2005; Szanyi et al., 2005; Yi et al., 2007; Yi and Szanyi, 2008)**. The experimental conditions (temperature, concentrations) and surfaces (catalytic materials) are quite different from the atmospheric conditions. Some rules summarized from dark reactions may not be suitable for the current photoreactions. We have concluded the nitrite oxidation process under dark condition **(line 204-206)**. '*The nitrite would react with another surface nitrite in a Langmuir-Hinshelwood mechanism (R.3) or gaseous $NO_2$ in an Eley-Rideal mechanism (R.4) to form nitrate in the absence of illumination* **(Tang et al., 2018; Underwood et al., 1999; Wu et al., 2013)**.' By the way, we found some knowledge on FT-IR analysis meaningful and cited these papers in the revised manuscript **(line 181-183)**. '*Bridging bidentate nitrate can be recognized by the shoulder peak at 1602 $cm^{-1}$* **(Du et al., 2019; Goodman et al., 1998; Sun et al., 2016; Szanyi et al., 2005; Yi et al., 2007)**.'

The photoinduced oxidation of nitrite intermediates has been thoroughly investigated. Experiments on $Al_2O_3$ have been added in section 3.2 of the new manuscript. The nitrite on $Al_2O_3$ is positively associated with nitrate products under various light intensities, resulting mainly from the photolysis process. In contrast, the association between nitrite and nitrate appears to be negative, indicating the significant photoinduced oxidation of nitrite intermediates. DRIFTS experiments have been added as well to confirm the effect. Noticeably, the 'nitrite-nitrate' association on authentic dust (Kaolinite) is negative as well, indicating that the photoinduced nitrite oxidation is much more significant than the photolysis process in the balance of diverse nitrogen species. More

details have been discussed in Section 3.2 and 3.4 of the revised manuscript.

Conclusions are made that are not supported by the results and overstate the importance of the results. For example, starting on line 297 the authors state, "*Generally, higher NO₂ concentrations suggest broader influence scope of illumination. Hence, current serious NO2 pollution may increase the participation of solar irradiation in the formation of secondary aerosols.*" It is not clear how this work actually addresses formation of secondary aerosol since most of the work looked at NO2 chemistry on $TiO_2$ and a rather limited study of nitrite and nitrate levels in ambient aerosols.

In another example, on line 322, the authors state, "*However, the nitrate formation on mineral dust is found to be more dependent on weak sunlight, indicated that photochemistry processes are still crucial in heavy haze.*" They further state as one of the main conclusions of the paper is that nitrate formation on mineral dust is "*more dependent on weak sunlight.*" Unfortunately, it was unclear to me how the authors can extract this conclusion from their data. It is well known that nitrate formation is driven by photochemical oxidation that converts NO2 to nitric acid/nitrate in a radical termination step. This photochemical pathway is directly proportional to light intensity, which drives both OH and NOx production rates. The nitrate formation rate on TiO2 is also shown to increase with light intensity as shown in Figure 1G. The authors also suggest there is a "*nitrite burst accompanied by low nitrate concentrations*" that has not been considered by the atmospheric community. It is not clear what burst they are referring to, as this part of the discussion was quite unclear. The conclusion appears to comes from the correlations observed in Figure 3, but those observations are expected and can be interpreted using known chemistry. For example the positive correlation between nitrite and nitrate at evening/night is expected since both stem from adsorption of N2O5 and NO2 to particles, while a negative correlation is expected during the day since high photon flux generates high concentrations of more photochemically stable nitric acid/nitrate, but effectively photolyzes nitrite.

Answer:

Thanks for the valuable comments on conclusions and we have changed this section based on the revised discussion section. The current conclusion section contains four parts:

(1) the summary of this study,

(2) the uneven illumination effect and the associations between nitrite and nitrate species on $TiO_2$

and authentic dust (Kaolinite)

(3)   the associations between atmospheric nitrite and nitrate

(4)   the importance of Monte Carlo simulation in the assessment of kinetics.

These four parts have contained the main contents of the revised manuscript. The conclusion section has been thoroughly improved in the revised manuscript **(line 445-473)**.

*'The heterogeneous reaction of $NO_2$ on $TiO_2$ and Kaolinite particles was investigated by means of in-situ DRIFTS experiments and ex-situ flow reactions, respectively. Photochemistry has conventionally been considered inapparent during haze events due to the weak sunlight near the ground caused by low visibility. In this work, light intensity is taken into consideration as an important index. For better illustration, size-segregated aerosol samples were collected in the absence and presence of sunlight.*

*Water-solvated nitrate is the main product on $TiO_2$ surfaces, followed by other species influenced by illumination conditions. The nitrate formation rate is enhanced by simulated solar irradiation and closely related to the light intensity. The nitrate formation is sensitive to the variation of light intensity under weak illumination, while tends to be insensitive under strong irradiation. That is, the uptake coefficient for $NO_2$ adsorption under moderate sunlight is close to that under strong irradiation. The oxidation of nitrite intermediates can be observed under dark condition and is promoted by the appearance of light and the elevation of the intensity. Hence, a significant negative correlation exists between nitrate and nitrite products. Similar uneven promotion effect and negative dependence occur on authentic particles (Kaolinite) as well. Based on the photo-electrochemical (PEC) tests and concentration dependence experiments, these illumination effects can be explained by the excess $NO_2$ adsorption under weak illumination while the sufficient photoinduced active species under strong irradiation.*

*The negative association between atmospheric nitrite and nitrate can be found in the presence of sunlight, along with the positive correlation in the absence of sunlight. The negative dependence is mainly attributed to the photoinduced oxidation of nitrite intermediates, suggesting illumination an inducing factor in the atmospheric nitrogen cycling. Nitrate pollution has got much concern recently, while little attention has been paid to the high nitrite concentrations accompanied by low nitrate levels. Nitrite may induce adverse health risk for its close association with various cancer cases* **(Zhang et al., 2018)**. *Compared to the nitrate-rich aerosols under strong irradiation, the nitrite-rich aerosols under faint sunlight may be more harmful to human health.*

*Actually, we discussed the γ-values based on the averaged experimental results while gave little care to various measurement errors. Compared to the arithmetic mean results, the percentile γ-values estimated by Monte Carlo simulation could be more suitable for modelling studies due to the differences between real atmosphere and the simulated laboratory condition. Furthermore, sensitivity analysis is helpful in explaining the determining factors involved in the assessment of uptake capacity. Generally, statistical simulation brings about more accurate evaluation and provides opportunities to explain the model discrepancy for secondary aerosols. '*

In summary, The paper is poorly organized and written; there is no experimental evidence that the ambient aerosol N chemistry is driven by or can be linked to TiO2 (anatase) chemistry; the TiO2 study is not conducted under atmospherically relevant conditions on a substrate that is not atmospherically relevant; and some of the main mechanisms proposed are either inaccurate or lack experimental support and can rather be explained using well-established chemistry. For this reason, I do am unable to recommend this manuscript for further consideration. Moving forward, I would suggest that the authors place all of the TiO2 data (which appears to be of good quality) into a concise manuscript (break the multi-panel figures up into more digestible figures) focused on the topic of photocatalytic reactions of N on TiO2 surfaces, which could be submitted to a more specialized journal in the area of catalysis or environmental remediation.

Answer:

We have noticed some probable shortcomings in the original manuscript and then made major revisions based on your comments.

**Firstly, the particles.**

As requested, in the revised manuscript we have provided a direct experimental evidence that the ambient aerosol N chemistry is driven by and can be linked to $TiO_2$ (anatase) chemistry. The association between nitrite and nitrate products is positive on $Al_2O_3$ while appears to be negative on $TiO_2$. The positive dependence on $Al_2O_3$ can be explained by the photolysis of nitrogen compounds, whereas the negative correlation on $TiO_2$ is caused by the photoinduced oxidation of nitrite intermediates, which can be observed by DRIFTS spectra as well. This suggests that the aerosol N chemistry observed in this work is directly driven by $TiO_2$. We have added the results on $Al_2O_3$ in the revised manuscript **(section 3.2)**. Based on these findings, photoactive mineral dusts may play

an essential role in atmospheric nitrogen chemistry. In atmospheric photoreactions, the effects of $TiO_2$ or $Fe_2O_3$ is quite different from those of non-photoactive components.

Furthermore, the latest results on authentic dust (Kaolinite) have been added **(section 2.3 and 3.4)**. We have introduced this point in the revised manuscript **(line 324-326)**. '*Although being ubiquitous in natural environment, $TiO_2$ is in representative of single-component mineral dust. Kaolinite, comprising $SiO_2$, $Al_2O_3$, $Fe_2O_3$, $TiO_2$ and other components **(Table S1)**, is a more typical authentic mineral dust.*' In brief, the uneven promotion effect on nitrate formation can be observed on Kaolinite as well. More importantly, the association between nitrite and nitrate products appears to be negative on Kaolinite, in consistent with that on $TiO_2$, indicating that the photoinduced oxidation of nitrite intermediates makes great contributions to the nitrate production on authentic dust. Hence, the illumination effects found on $TiO_2$ exist in the atmospheric environment. In general, $TiO_2$ could be regarded as the surrogate of mineral dust when discussing the atmospheric photoreactions.

**Secondly, the atmospheric condition.**

In the original manuscript, the $NO_2$ concentration for $TiO_2$ particles was relatively high ($3.77 \times 10^{14}$ molecules $cm^{-3}$). However, for the new sections on Kaolinite particles, we adopted a much lower $NO_2$ mixing ratio ($2.21 \times 10^{13}$ molecules $cm^{-3}$) to get close to the real atmospheric condition. Simultaneously, because the low $NO_2$ concentration hinders the product observation via *in-situ* DRIFTS, a new-style quartz flow reactor has been designed and employed in the Kaolinite-related experiments. We believe that the experiments under lower $NO_2$ concentrations are helpful in explaining the illumination effects.

**Thirdly, the mechanism for $NO_2$ adsorption.**

$N_2O_4$ was not observed by *in-situ* DRIFTS, and may exist in the very initial stage of the reaction. We have added the discussions on $N_2O_4$ and the possible reaction processes triggered by it in the revised manuscript **(line 224-229)**.

Furthermore, photolysis of nitrogen compounds was not thoroughly discussed in the original manuscript. We have added the discussion on the photolysis process in the revised manuscript **(line 210-223)**. Additionally, the photolysis process is a nice explanation on the negative association between nitrite and nitrate aerosols, and we have added this point inti the revised manuscript **(line 380-386)**.

The mechanism on the photoinduced heterogeneous reaction of $NO_2$ on mineral dust has been widely reported. However, the **uneven promotion effect on nitrate formation** under illumination was seldom studied. Meanwhile, **the knowledge of atmospheric nitrite**, **the associations between nitrite and nitrate aerosols**, and the **photoinduced oxidation of nitrite intermediates** were scarcely reported in the earlier studies. These issues have been thoroughly investigated in the revised manuscript. Hence, this paper is novel in its community and would provide more opportunities for the further studies. We have emphasized these points in the introduction section **(line 66-74)**. *'Most remarkable studies concerned the photocatalytic effects under ultraviolet light while the intensity dependence under simulated solar irradiation was rarely explored* **(Dupart et al., 2014; Guan et al., 2014; Li et al., 2010; Shang et al., 2010)***. Some researchers* **(El Zein and Bedjanian, 2012)** *noticed the uneven illumination effect on the heterogeneous reaction of $NO_2$ on $TiO_2$ particles while ignored the reaction process on authentic dust and the reaction mechanism behind the variation. Furthermore, nitrite owns an important position in atmospheric processes for its frequent appearance and great contributions to aerosol toxicity. However, there is little information available in literature about the pollution characteristics or reaction pathways of nitrite aerosols. Generally, how the illumination influences the uptake capacity and product species are questions urgently needs solving.'*

We believe that the manuscript has been greatly improved based on the comments from the reviewers and editor. Generally, extra tests on $Al_2O_3$, experiments on authentic dust, lower $NO_2$ concentration, and more mechanism discussions have been added into the revised manuscript. We are happy to listen to any further suggestions from the reviewers.

Specific Comments: Too many figures are shown in Figure 1. These figures can easily be broken up in a way to make a separate manuscript on its own (see above comment).

Answer:

In the revised manuscript, we have divided the original Figure 1 into two parts. The current Figure 1 contains the DRIFTS spectra on $TiO_2$ particles. The IC measurements for the nitrite and nitrate products on $TiO_2$ and $Al_2O_3$ particles under diverse light intensities have been presented in the current Figure 2.

Additionally, in order to make the readers better understand Figure 1, we have explained the

general content of this figure in the beginning of the section 3.1 **(line 174-176)**. '*Figure 1 presents the product analysis for dark reaction* **(Figure 1a-c)** *and illumination process* **(Figure 1d-f)** *by means of DRIFTS spectra. Gaussian curve-fitting procedure was employed to deconvolute the overlapping bands.*'

line 62: The authors state that some researchers have studied the effect of radiation on $NO_2$ uptake on $TiO_2$ but have ignored the reaction mechanism behind the trends in reactivity. However, after reading the papers cited and some that are not cited, I disagree.

Answer:

The mechanism of $NO_2$ adsorption on mineral dust has been exhaustively studied. However, little attention has been paid to the heterogeneous reaction influenced by light intensity, especially weak irradiation during heavy haze events. Some researchers noticed the uneven illumination effect on the heterogeneous reaction of $NO_2$ on $TiO_2$ particles under UV light while ignored the reaction process on authentic dust and the reaction mechanism behind the variation.

We have emphasized these details in the introduction section **(line 66-70)**. '*Most remarkable studies concerned the photocatalytic effects under ultraviolet light while the intensity dependence under simulated solar irradiation was rarely explored* **(Dupart et al., 2014; Guan et al., 2014; Li et al., 2010; Shang et al., 2010)**. *Some researchers* **(El Zein and Bedjanian, 2012)** *noticed the uneven illumination effect on the heterogeneous reaction of $NO_2$ on $TiO_2$ particles while ignored the reaction process on authentic dust and the reaction mechanism behind the variation.*'

line 229-230. In figure 2c, the authors fit a polynomial function to the experimental data collected for NO2 uptake coefficients plotted as a function of light intensity. They then provide an inset showing a near perfect correlation between experimental uptake coefficients and those making up the line they fit to it. This is unnecessary. I recommend omitting the inset and simply reporting the R-squared value for the polynomial fit. 237-238: The polynomial fits lack any physical meaning and are only applicable to the substrate and conditions of the experiment, which are not necessarily environmentally applicable (see above).

Answer:

The regression analysis is aimed at finding an accurate formula fitting the obtained curve. The

polynomial, inspired by previous reports, was constructed for the better estimation of $d\gamma_{geo}/dI$ values **(El Zein and Bedjanian, 2012)**. We have emphasized these details in the revised manuscript *(line 308-310)*. '*Based on this, the $d\gamma_{geo}/dI$ values can be obtained by derivation to distinguish the illumination effect varying with light intensity.*' Figure 4(c) makes it easier for readers to understand the generation of $d\gamma_{geo}/dI$ values.

For the similar data analysis on Kaolinite experimental results, we ignored the process and presented uptake coefficients and $d\gamma_{geo}/dI$ values in the same figure (Figure 3c). Meanwhile, we provided only the R-square value in discussing the situation for Kaolinite in the revised manuscript **(line 333-334)**. '*Based on regression analysis (R²=0.994), the $d\gamma_{geo}/dI$ values were estimated by derivation to distinguish the promotion effect influenced by light intensity.*'

Figure 3: It is entirely unclear what the rightmost column is depicting. What do the authors mean by "nitrite classification?" What do the color codes mean?

Answer:

The rightmost column exhibits the correlations between daily nitrite and nitrate concentrations in (c) coarse mode, (f) droplet mode, (i) condensation mode and (l) full-size particles. The insets present the associations between nitrite and nitrate based on the classification of nitrite levels.

The 'nitrite classification' was carried out to divide the aerosol samples into several groups according to the nitrite concentration for better analysis. We have explained this process in the revised manuscript **(line 359-360)**. '*For better comparison, the 26 daily samples were classified into several groups according to the nitrite concentrations.*'

The color of the rightmost column was designed for the classification of nitrite concentration. We have added the color scale in the revised figure.

**Anonymous Referee #3**

This is a new contribution aiming at understanding the effect of mild illumination on mineral dust proxies and linking those observations with a limited set of environmental data. The claim having motivated this investigation can be found in the fact that previous studies have not seriously discussed the photochemistry under weak sunlight during haze conditions, and have thus ignored some underlying processes. I would certainly agree on such a statement, that I found quite interesting. However, I'm not fully convinced by the current version of the manuscript that would need significant revision to convey a better and stronger elaborated message.

First of all, I do need to admit that I found it difficult to read this manuscript which contains a series of awkward sentences, such: "The growth of nitrate formation rate tends to be slow after the initial fast with increasing light intensity. For example, the geometric uptake coefficient ($\gamma_{geo}$) under 30.5 mW/cm$^2$ ($5.72\times10^{-6}$) has exceeded the 50 % of that under 160 mW/cm$^2$ ($1.13\times10^{-5}$). This case can be explained by the excess $NO_2$ adsorption under weak illumination while the excess photoinduced active species under strong irradiation. Being negatively associated with nitrate (R2=0.748, P<0.01), nitrite acts as the intermediate and decreases with: : :". This clearly weakens the content of this manuscript.

Answer:

Thanks a lot for your valuable comments, which greatly improved this work.

We have made major revisions on the abstract section. In the revised manuscript the paragraph has been changed as follows.

*'Despite the increased awareness of heterogeneous reaction on mineral dust, the knowledge of how the intensity of solar irradiation influences the photochemistry activity remains a crucially important part in atmospheric research. Relevant studies have not seriously discussed the photochemistry under weak sunlight during haze, and thus ignored some underlying pollution and toxicity. Here, we investigated the heterogeneous formation of nitrate and nitrite on mineral dust under various illumination conditions by laboratory experiments and field observations. Observed by in-situ diffuse reflectance infrared Fourier transform spectroscopy (DRIFTS), water-solvated nitrate was the main product on $TiO_2$ surfaces, followed by other species varying with illumination condition. The nitrate formation rate grows rapidly from the dark to faint sunlight, while becomes*

*insensitive to light intensity variation under strong irradiation. For example, the geometric uptake coefficient ($\gamma_{geo}$) under 30.5 mW/cm$^2$ (5.72×10$^{-6}$) has exceeded the 50 % of that under 160 mW/cm$^2$ (1.13×10$^{-5}$). Additionally, being negatively associated with nitrate ($R^2$=0.748, p<0.01), nitrite acts as the intermediate and decreases with increasing light intensity via oxidation pathways. Heterogeneous reaction on authentic dust (Kaolinite) exhibits the similar uneven promotion effect and photoinduced negative association, which can be explained by the excess $NO_2$ adsorption under weak illumination while the sufficient photoinduced active species under strong irradiation based on the photo-electrochemical tests and concentration dependence experiments. Similar negative dependence appears in coarse particles collected during daytime ($R^2$=0.834, p<0.05), accompanied by the positive association during nighttime ($R^2$=0.632, P<0.05), suggesting illumination a substantial role in the atmospheric nitrogen cycling. Overall, for the nitrate formation, the conspicuous response under slight illumination offers opportunities to explain the secondary aerosol burst during haze episodes with weak irradiation. Additionally, high nitrite levels accompanied by low nitrate concentrations may induce great health risk which was previously neglected. Further, Monte Carlo simulation coupled with sensitivity analysis may provide a new insight in the estimations of kinetics parameters for atmospheric modelling studies.'*

Then the link being made with the ambient measurements and the test on $TiO_2$ particles is far from being obvious, and would certainly need to more elaborated to make a stronger case.

Answer:

Some revisions have been made on the manuscript to support the ambient measurements.

**Firstly**, $Al_2O_3$ has been selected as the surrogate of atmospheric non-photoactive components. The DRIFTS experiments on $TiO_2$ have been repeated on $Al_2O_3$. For the links between nitrite and nitrate products on mineral dust surrogates, the association appears to be positive on $Al_2O_3$, whereas appears to be negative on $TiO_2$, resulting mainly from the photolysis process and the photoinduced oxidation of nitrite intermediates, respectively. The comparisons between $Al_2O_3$ and $TiO_2$ provide strong evidences to support the oxidation of nitrite species under irradiation, which was scarcely mentioned in previous studies. The photoinduced oxidation of nitrite species can be observed by DRIFTS as well. We have added these points into the section 3.2 of the revised manuscript.

**Secondly**, although being ubiquitous in natural environment, $TiO_2$ is in representative of

single-component mineral dust. Kaolinite, comprising $SiO_2$, $Al_2O_3$, $Fe_2O_3$, $TiO_2$ and other components, is a more typical authentic mineral dust. The association between nitrite and nitrate under diverse light intensities is negative on Kaolinite particles, in consistent with that on $TiO_2$ while differs from the positive trend on $Al_2O_3$. Despite the small shares of photoactive components (mainly $TiO_2$ and $Fe_2O_3$) on the Kaolinite particles, the photoinduced oxidation of nitrite intermediate seems to be much more significant compared to the photolysis processes. Hence, atmospheric photoactive components (e.g. $TiO_2$, $Fe_2O_3$) plays an important role in the mineral dust when discussing the photoinduced reactions.

**Finally**, more discussions have been made on the ambient measurements. For example, the associations in droplet and condensation modes have been summarized **(line 364-369)**. '*The associations in droplet (Figure 6d-f) and condensation (Figure 6g-i) modes are comparable with those on coarse mode. Inspired by the discussion above, the positive associations during nighttime are attributed to the $NO_2$ disproportionation process on particle surfaces, in suspended droplets, or during the collisions with water vapour. In the presence of solar irradiation, the negative dependence can be explained by the oxidation of nitrite intermediates promoted by various active species in different media.*' Additionally, the photolysis of nitrogen components is helpful in explaining the negative associations **(line 380-386)**. '*It is universally acknowledged that atmospheric nitrate and nitrite may undergo photolysis under acidic conditions. These two processes (photolysis of nitrogen compounds and oxidation of nitrite) may contribute to the negative associations in different aspects. Yet, the nitrite oxidation seems to be more easily to take place because some photoactive components in the atmosphere can be excited by visible light, while the photolysis mainly occurs under ultraviolet light. The exact contributions of these processes to the negative correlation are still not very clear. However, the photolysis of atmospheric nitrate and nitrite is a strong evidence in explaining the low nitrite levels accompanied by slight nitrate pollution in the presence of sunlight.*'

Most of the data presented have been made at quite high $NO_2$ concentrations (tens of ppm), corresponding to a concentration regime where $N_2O_4$ is known to be a significant intermediate for NOx conversion on surfaces. While such concentrations are inherent to the DRIFT technique, one can still wonder how this could affect the findings reported here. It is stated that "Since no saturation

effects were observed in the DRIFTS experiments, the $NO_2$ adsorption rate can be regarded as constant.", but this is strong contrast with previous studies dedicated to NO2 heterogeneous chemistry. Could that be to the involvement of N2O4?

Answer:

$N_2O_4$ was not observed in the DRIFTS spectra. It is deduced that $N_2O_4$ formed in the initial stage of the whole reaction and then fade away during oxidation steps. Previous studies reported the oxidation of S(IV) species to S(VI) compounds along with the consumption of $N_2O_4$. Hence, $N_2O_4$ is considered as an promotor during the oxidation of nitrite. We have added some discussions in the revised manuscript **(line 224-229)**. '*Different with previous studies, the dimer of $NO_2$, namely $N_2O_4$, was not observed in these spectra around 1745 and 1295 $cm^{-1}$. It has been confirmed that $N_2O_4$ could isomerize and autoionize to $NO^+NO_3^-$, and then react with adsorbed water to generate $HNO_3$ and HONO* **(Syomin and Finlayson-Pitts, 2003)**. *Meanwhile, $N_2O_4$ may exist in the initial stage of the whole reaction and be consumed immediately along with the oxidation of nitrite species. Liu et al., investigated the consumption of $N_2O_4$ during the oxidation of adsorbed $SO_2$* **(Liu et al., 2012)**. *Herein, the proposed process can be described by (R.6-7).*'

The original sentence 'Since no saturation effects were observed in the DRIFTS experiments, the $NO_2$ adsorption rate can be regarded as constant.' was aimed at explaining the continuous nitrate formation during each DRIFTS test. Based on this, we started the discussion on the balance between $NO_2$ adsorption and PAS generation, which results in the uneven promotion effect under illumination. $N_2O_4$ was not observed after the initial reaction stage, and was thus not considered in mechanism discussion. Additionally, the original sentence has been changed in the revised manuscript for better understanding **(line 403-404)**. '*Since no saturation effects were observed in each in-situ experiment, the $NO_2$ adsorption rate can be regarded as constant.*'

In TiO2 driven photocatalysis, it is known that the conversion rate is linear function of light intensity at low intensity, and then levels off at high power to reach steady state conditions. While this seems to be also observed here (inset of figure 2a) but not really discussed nor mentioned, why? In addition, one could argue that the light intensities used here are far from being small and I would not consider this as being photochemistry under mild conditions.

Answer:

Most TiO$_2$ photocatalytic experiments were conducted under UV light. Herein, TiO$_2$ is considered as the surrogate of mineral dust, and exposed to NO$_2$ under simulated solar irradiation. This study is aimed at exploring the heterogeneous reaction of NO$_2$ on mineral dust in the atmospheric environment. The nitrate formation rate grows rapidly from the dark to faint sunlight, while becomes insensitive to light variation under strong irradiation. This uneven promoting effect under illumination can be explained by the excess NO$_2$ adsorption under weak illumination while the sufficient photoinduced active species under strong irradiation. We have emphasized the difference between this study and others in the introduction section **(line 66-68)**. *'Most remarkable studies concerned the photocatalytic effects under ultraviolet light while the intensity dependence under simulated solar irradiation was rarely explored* **(Dupart et al., 2014; Guan et al., 2014; Li et al., 2010; Shang et al., 2010)**.*'* Meanwhile, we have explained the uneven illumination effect, which acts as one of the most important findings in section 3.6 of the revised manuscript.

The light intensities for TiO$_2$ particles are 0-160 mW/cm$^2$. At the global AM (air mass) 1.5 G (global) condition, 1 sun is defined as equal to 100 mW/cm$^2$ of irradiance (a standard used in solar cell research). We adopted some high light intensities on TiO$_2$ to explore the general rules of the illumination effects. For the authentic dust Kaolinite, we designed all tests under the light intensities of 0-100 mW/cm$^2$ with a much lower NO$_2$ concentration (*2.21×10$^{13}$ molecules cm$^{-3}$*). We have introduced this detail in the method section **(line 113-114)**. *'In each test, 50 mg Kaolinite was put into the chamber, followed by the exposure to 2.21×10$^{13}$ molecules cm$^{-3}$ NO$_2$ under diverse light intensities (0, 10, 40, 70, and 100 mW/cm$^2$)'* Additionally, more details have been included in the section 2.3 of the revised manuscript. We believe the added experiments on Kaolinite particles are providing more evidences to support the conclusions.

One of the key finding here is associated with the nitrite levels between sustained (or more important) under mild illumination, but the section "mechanism" does not provide a real explanation for that. You should try to define what you mean with cumulative uptake coefficients.

Answer:

Based on the photo-electrochemical (PEC) tests and concentration dependence experiments, the uneven promotion effect on nitrate formation and the negative association between nitrite and nitrate can be explained by the excess NO$_2$ adsorption under weak illumination while the sufficient

photoinduced active species under strong irradiation. We have emphasized the nitrite oxidation processes under different illumination conditions in the revised manuscript **(line 430-436)**. '*The balance between $NO_2$ adsorption and PAS formation influenced by light intensity can also be used to explain the negative association between nitrate and nitrite. Limited PAS participate in the oxidation of excess $NO_2$ under weak illumination, and thus make space for the nitrite formation via disproportionation process. Under strong irradiation, there are still sufficient PAS involved in the nitrite oxidation after the photochemical conversion of limited $NO_2$. For the oxidation of nitrite intermediates, the main promoters are $NO_2$ and other nitrite species under dark condition or weak illumination, while turns to be PAS under stronger irradiation. Hence, nitrite unevenly decreases with increasing light intensity, and exhibits a negative association with nitrate.*' Generally, in the mechanism section, we explained the illuminations effects based on the PEC tests and the concentration dependence experiments. We have explained this in the beginning of the mechanism section in the revised manuscript **(Line 398-399)**. '*Detailed reaction pathways concerning the heterogeneous reaction of $NO_2$ on mineral dust have been included in previous sections.*' Meanwhile, the photolysis of nitrate and nitrite may also be used to explain the negative dependence between nitrite and nitrate. Hence, we changed the title of section 3.6 to '*Proposed mechanism*'.

Monte Carlo simulation was implemented to quantify the uncertainty and its impact on the kinetics assessment. Each independent variable was determined via five or more replication measurements and assumed to be normally distributed in the simulation. We performed independent runs at 1500, 3000, 5000, and 10000 iterations with each parameter. The results showed that 5000 iterations are sufficient to ensure the stability of the results. Hence, the phrase 'cumulative uptake coefficients', which was not mentioned in the manuscript, includes 5000 calculated results. Instead of the cumulative uptake coefficients, we used the percentile values for each case based on the cumulative probability of uptake coefficients (summarized in Table S3, Table S4). Noticeably, arithmetic mean results are frequently used in this paper for better comparison with other papers, and we have emphasized this detail in the revised manuscript **(line 154-155)**. '*Arithmetic mean values are used in the following comparisons for better understanding.*'

A few chemical reactions are described in a simplified i.e., wrong way with unbalanced stoichiometry. This should be avoided.

Answer:

Numerous reaction processes were included when discussing the heterogeneous reaction on mineral dust. Some important reactions were summarized in the manuscript in a brief form, followed by some others listed in the Supporting Information. We have emphasized it in the revised manuscript **(line 206-207)**. *'The reactions R1 and R2 are presented in a brief way, followed by detailed processes listed in Section S8.'*

I did found Figure 3 quite difficult to understand and I am finally unsure about the message the authors wants to convey with this illustration. This maybe needs to be better discussed.

Answer:

The Figure 3 (Figure 6 in the revised manuscript) illustrates the associations between nitrite and nitrate aerosols varying with particle size (coarse mode, droplet mode, condensation mode, and full-size) and collection period (nighttime, daytime, and whole day). The insets present the associations based on the classification of nitrite levels. The classification has been explained in the revised manuscript **(line 359-360)**. '*For better comparison, 26 daily samples were classified into several groups according to the nitrite concentrations.*' Additionally, the color in the rightmost column reveals the classification of nitrite concentration. We have added the color scale in the revised figure.

Overall, this figure provides a strong evidence to the nitrite oxidation in the presence of illumination. We have summarized these results in the revised manuscript **(line 387-389)**. '*Generally, atmospheric nitrite is positively correlated with nitrate in the absence of irradiation, whereas presents negative association with nitrate during daytime. The dependence is significant for coarse particles while turns to be inconspicuous in droplet mode and condensation mode.*'

**References**

Balajka, J., Hines, M.A., DeBenedetti, W.J.I., Komora, M., Pavelec, J., Schmid, M., Diebold, U.: High-affinity adsorption leads to molecularly ordered interfaces on $TiO_2$ in air and solution, Science, 361, 786-789, doi: 10.1126/science.aat6752, 2018.

Ballari, M.M., Hunger, M., Hüsken, G., Brouwers, H.J.H.: $NO_x$ photocatalytic degradation employing concrete pavement containing titanium dioxide, Applied Catalysis B: Environmental, 95, 245-254, doi: 10.1016/j.apcatb.2010.01.002, 2010.

Ballari, M.M., Yu, Q.L., Brouwers, H.J.H.: Experimental study of the NO and $NO_2$ degradation by photocatalytically active concrete, Catal. Today, 161, 175-180, doi: 10.1016/j.cattod.2010.09.028, 2011.

Benedict, K.B., McFall, A.S., Anastasio, C.: Quantum Yield of Nitrite from the Photolysis of Aqueous Nitrate above 300 nm, Environ. Sci. Technol., 51, 4387-4395, doi: 10.1021/acs.est.6b06370, 2017.

Chen, H., Nanayakkara, C.E., Grassian, V.H.: Titanium dioxide photocatalysis in atmospheric chemistry, Chem. Rev., 112, 5919-5948, doi: 10.1021/cr3002092, 2012.

Chen, W., Zhu, T.: Formation of Nitroanthracene and Anthraquinone from the Heterogeneous Reaction Between $NO_2$ and Anthracene Adsorbed on NaCl Particles, Environ. Sci. Technol., 48, 8671-8678, doi:2014.

Chen, Y., Tong, S., Wang, J., Peng, C., Ge, M., Xie, X., Sun, J.: Effect of Titanium Dioxide on Secondary Organic Aerosol Formation, Environ. Sci. Technol., doi: 10.1021/acs.est.8b02466, 2018.

Desikusumastuti, A., Staudt, T., Grönbeck, H., Libuda, J.: Identifying surface species by vibrational spectroscopy: Bridging vs monodentate nitrates, J. Catal., 255, 127-133, doi: 10.1016/j.jcat.2008.01.019, 2008.

Du, C., Kong, L., Zhanzakova, A., Tong, S., Yang, X., Wang, L., Fu, H., Cheng, T., Chen, J., Zhang, S.: Impact of adsorbed nitrate on the heterogeneous conversion of $SO_2$ on α-$Fe_2O_3$ in the absence and presence of simulated solar irradiation, Sci. Total Environ., 649, 1393-1402, doi: 10.1016/j.scitotenv.2018.08.295, 2019.

Dupart, Y., Fine, L., D'Anna, B., George, C.: Heterogeneous uptake of $NO_2$ on Arizona Test Dust under UV-A irradiation: An aerosol flow tube study, Aeolian Res., 15, 45-51, doi: org/10.1016/j.aeolia.2013.10.001, 2014.

El Zein, A., Bedjanian, Y.: Interaction of $NO_2$ with $TiO_2$ surface under UV irradiation: measurements of the uptake coefficient, Atmos. Chem. Phys., 12, 1013-1020, doi: 10.5194/acp-12-1013-2012, 2012.

Fu, H., Cwiertny, D.M., Carmichael, G.R., Scherer, M.M., Grassian, V.H.: Photoreductive dissolution of Fe-containing mineral dust particles in acidic media, Journal of Geophysical Research, 115, D11304, doi: 10.1029/2009JD012702, 2010.

Gen, M., Zhang, R., Huang, D.D., Li, Y., Chan, C.K.: Heterogeneous Oxidation of $SO_2$ in Sulfate Production during Nitrate Photolysis at 300 nm: Effect of pH, Relative Humidity, Irradiation Intensity, and the Presence of Organic Compounds, Environ. Sci. Technol., doi: 10.1021/acs.est.9b01623, 2019.

Goodman, A.L., Miller, T.M., Grassian, V.H.: Heterogeneous reactions of $NO_2$ on NaCl and $Al_2O_3$ particles, Journal of Vacuum Science & Technology A: Vacuum, Surfaces, and Films, 16, 2585-2590, doi: 10.1116/1.581386, 1998.

Guan, C., Li, X., Luo, Y., Huang, Z.: Heterogeneous Reaction of $NO_2$ on α-$Al_2O_3$ in the Dark and Simulated Sunlight, J. Phys. Chem. A, 118, 6999-7006, doi: 10.1021/jp503017k, 2014.

Gustafsson, R.J., Orlov, A., Griffiths, P.T., Cox, R.A., Lambert, R.M.: Reduction of $NO_2$ to nitrous acid on illuminated titanium dioxide aerosol surfaces: implications for photocatalysis and atmospheric

chemistry, Chem. Commun., 0, 3936-3938, doi: 10.1039/B609005B, 2006.

Hofer, U.: Illuminating the importance of nitrite oxidation, Nat. Rev. Microbiol., 16, 65, doi: 10.1038/nrmicro.2017.165, 2018.

Huang, L., Zhao, Y., Li, H., Chen, Z.: Kinetics of Heterogeneous Reaction of Sulfur Dioxide on Authentic Mineral Dust: Effects of Relative Humidity and Hydrogen Peroxide, Environ. Sci. Technol., 49, 10797-10805, doi: 10.1021/acs.est.5b03930, 2015.

Kebede, M.A., Varner, M.E., Scharko, N.K., Gerber, R.B., Raff, J.D.: Photooxidation of Ammonia on $TiO_2$ as a Source of NO and $NO_2$ under Atmospheric Conditions, J. Am. Chem. Soc., 135, 8606-8615, doi: 10.1021/ja401846x, 2013.

Kim, D., Lee, J., Ryu, J., Kim, K., Choi, W.: Arsenite Oxidation Initiated by the UV Photolysis of Nitrite and Nitrate, Environ. Sci. Technol., 48, 4030-4037, doi: 10.1021/es500001q, 2014.

Langridge, J.M., Gustafsson, R.J., Griffiths, P.T., Cox, R.A., Lambert, R.M., Jones, R.L.: Solar driven nitrous acid formation on building material surfaces containing titanium dioxide: A concern for air quality in urban areas? Atmos. Environ., 43, 5128-5131, doi: 10.1016/j.atmosenv.2009.06.046, 2009.

Lesko, D.M.B., Coddens, E.M., Swomley, H.D., Welch, R.M., Borgatta, J., Navea, J.G.: Photochemistry of nitrate chemisorbed on various metal oxide surfaces, Phys. Chem. Chem. Phys., 17, 20775-20785, doi: 10.1039/C5CP02903A, 2015.

Li, H.J., Zhu, T., Zhao, D.F., Zhang, Z.F., Chen, Z.M.: Kinetics and mechanisms of heterogeneous reaction of $NO_2$ on $CaCO_3$ surfaces under dry and wet conditions, Atmos. Chem. Phys., 10, 463-474, doi: org/10.5194/acp-10-463-2010, 2010.

Li, J., Shang, J., Zhu, T.: Heterogeneous reactions of $SO_2$ on ZnO particle surfaces, Science China Chemistry, 54, 161-166, doi: org/10.1007/s11426-010-4167-9, 2010.

Li, L., Chen, Z.M., Zhang, Y.H., Zhu, T., Li, J.L., Ding, J.: Kinetics and mechanism of heterogeneous oxidation of sulfur dioxide by ozone on surface of calcium carbonate, Atmos. Chem. Phys., 6, 2453-2464, doi:2006.

Liu, C., Ma, Q., Liu, Y., Ma, J., He, H.: Synergistic reaction between $SO_2$ and $NO_2$ on mineral oxides: a potential formation pathway of sulfate aerosol, Phys. Chem. Chem. Phys., 14, 1668-1676, doi: 10.1039/C1CP22217A, 2012.

Liu, Y., Han, C., Ma, J., Bao, X., He, H.: Influence of relative humidity on heterogeneous kinetics of $NO_2$ on kaolin and hematite, Phys. Chem. Chem. Phys., 17, 19424-19431, doi: 10.1039/C5CP02223A, 2015.

Monge, M.E., George, C., D Anna, B., Doussin, J., Jammoul, A., Wang, J., Eyglunent, G., Solignac, G., Daële, V., Mellouki, A.: Ozone Formation from Illuminated Titanium Dioxide Surfaces, J. Am. Chem. Soc., 132, 8234-8235, doi: 10.1021/ja1018755, 2010.

Moon, D.R., Taverna, G.S., Anduix-Canto, C., Ingham, T., Chipperfield, M.P., Seakins, P.W., Baeza-Romero, M., Heard, D.E.: Heterogeneous reaction of $HO_2$ with airborne $TiO_2$ particles and its implication for climate change mitigation strategies, Atmos. Chem. Phys., 18, 327-338, doi: 10.5194/acp-18-327-2018, 2018.

Nanayakkara, C.E., Jayaweera, P.M., Rubasinghege, G., Baltrusaitis, J., Grassian, V.H.: Surface Photochemistry of Adsorbed Nitrate: The Role of Adsorbed Water in the Formation of Reduced Nitrogen Species on $\alpha$-$Fe_2O_3$ Particle Surfaces, The Journal of Physical Chemistry A, 118, 158-166, doi: 10.1021/jp409017m, 2014.

Nanayakkara, C.E., Larish, W.A., Grassian, V.H.: Titanium Dioxide Nanoparticle Surface Reactivity with Atmospheric Gases, $CO_2$, $SO_2$, and $NO_2$: Roles of Surface Hydroxyl Groups and Adsorbed Water

in the Formation and Stability of Adsorbed Products, The Journal of Physical Chemistry C, 118, 23011-23021, doi: 10.1021/jp504402z, 2014.

Ndour, M., Anna, B.D., George, C., Ka, O., Balkanski, Y., Kleffmann, J., Stemmler, K., Ammann, M.: Photoenhanced uptake of $NO_2$ on mineral dust: Laboratory experiments and model simulations, Geophys. Res. Lett., 35, L5812, doi: org/10.1029/2007GL032006, 2008.

Ndour, M., D'Anna, B., George, C., Ka, O., Balkanski, Y., Kleffmann, J., Stemmler, K., Ammann, M.: Photoenhanced uptake of $NO_2$ on mineral dust: Laboratory experiments and model simulations, Geophys. Res. Lett., 35, doi: 10.1029/2007GL032006, 2008.

Niu, H., Li, K., Chu, B., Su, W., Li, J.: Heterogeneous Reactions between Toluene and $NO_2$ on Mineral Particles under Simulated Atmospheric Conditions, Environ. Sci. Technol., 51, 9596-9604, doi: 10.1021/acs.est.7b00194, 2017.

Ozensoy, E., Peden, C.H.F., Szanyi, J.: $NO_2$ Adsorption on Ultrathin $\theta$-$Al_2O_3$ Films: Formation of Nitrite and Nitrate Species, The Journal of Physical Chemistry B, 109, 15977-15984, doi: 10.1021/jp052053e, 2005.

Park, J., Jang, M., Yu, Z.: Heterogeneous Photo-oxidation of $SO_2$ in the Presence of Two Different Mineral Dust Particles: Gobi and Arizona Dust, Environ. Sci. Technol., 51, 9605-9613, doi: 10.1021/acs.est.7b00588, 2017.

Ponczek, M., George, C.: Kinetics and product formation during the photooxidation of butanol on atmospheric mineral dust, Environ. Sci. Technol., 52, 5191-5198, doi: 10.1021/acs.est.7b06306, 2018.

Roca, M., Zahardis, J., Bone, J., El-Maazawi, M., Grassian, V.H.: 310 nm Irradiation of Atmospherically Relevant Concentrated Aqueous Nitrate Solutions: Nitrite Production and Quantum Yields, The Journal of Physical Chemistry A, 112, 13275-13281, doi: 10.1021/jp809017b, 2008.

Rubasinghege, G., Grassian, V.H.: Photochemistry of Adsorbed Nitrate on Aluminum Oxide Particle Surfaces, The Journal of Physical Chemistry A, 113, 7818-7825, doi: 10.1021/jp902252s, 2009.

Schuttlefield, J., Rubasinghege, G., El-Maazawi, M., Bone, J., Grassian, V.H.: Photochemistry of Adsorbed Nitrate, J. Am. Chem. Soc., 130, 12210-12211, doi: 10.1021/jp902252s, 2008.

Shang, J., Li, J., Zhu, T.: Heterogeneous reaction of $SO_2$ on $TiO_2$ particles, Science China Chemistry, 53, 2637-2643, doi: org/10.1007/s11426-010-4160-3, 2010.

Sun, X., Ji, Q., Jayakumar, A., Ward, B.B.: Dependence of nitrite oxidation on nitrite and oxygen in low-oxygen seawater, Geophys. Res. Lett., 44, 7883-7891, doi: 10.1002/2017GL074355, 2017.

Sun, Z., Kong, L., Ding, X., Du, C., Zhao, X., Chen, J., Fu, H., Yanga, X., Cheng, T.: The effects of acetaldehyde, glyoxal and acetic acid on the heterogeneous reaction of nitrogen dioxide on gamma-alumina, Phys. Chem. Chem. Phys., 18, 9367-9376, doi: 10.1039/C5CP05632B, 2016.

Sun, Z., Kong, L., Zhao, X., Ding, X., Fu, H., Cheng, T., Yang, X., Chen, J.: Effect of Formaldehyde on the Heterogeneous Reaction of Nitrogen Dioxide on $\gamma$-Alumina, The Journal of Physical Chemistry A, 119, 9317-9324, doi: 10.1021/acs.jpca.5b06632, 2015.

Syomin, D.A., Finlayson-Pitts, B.J.: HONO decomposition on borosilicate glass surfaces: implications for environmental chamber studies and field experiments, Phys. Chem. Chem. Phys., 5, 5236, doi: 10.1039/b309851f, 2003.

Szanyi, J., Kwak, J.H., Kim, D.H., Burton, S.D., Peden, C.H.F.: $NO_2$ Adsorption on $BaO/Al_2O_3$: The Nature of Nitrate Species, The Journal of Physical Chemistry B, 109, 27-29, doi: 10.1021/jp044993p, 2005.

Tan, F., Jing, B., Tong, S., Ge, M.: The effects of coexisting $Na_2SO_4$ on heterogeneous uptake of $NO_2$ on $CaCO_3$ particles at various RHs, Sci. Total Environ., 586, 930-938, doi:

org/10.1016/j.scitotenv.2017.02.072, 2017.

Tan, F., Tong, S., Jing, B., Hou, S., Liu, Q., Li, K., Zhang, Y., Ge, M.: Heterogeneous reactions of $NO_2$ with $CaCO_3$-$(NH_4)_2SO_4$ mixtures at different relative humidities, Atmos. Chem. Phys., 16, 8081-8093, doi: 10.5194/acp-16-8081-2016, 2016.

Tang, M., Keeble, J., Telford, P.J., Pope, F.D., Braesicke, P., Griffiths, P.T., Abraham, N.L., McGregor, J., Watson, I.M., Cox, R.A., Pyle, J.A., Kalberer, M.: Heterogeneous reaction of $ClONO_2$ with $TiO_2$ and $SiO_2$ aerosol particles: implications for stratospheric particle injection for climate engineering, Atmos. Chem. Phys., 16, 15397-15412, doi: 10.5194/acp-16-15397-2016, 2016.

Tang, S., Ma, L., Luo, M., Zhang, Z., Cao, X., Huang, Z., Xia, R., Qiu, Y., Feng, S., Zhang, P., Xia, C., Jin, Y., Xu, D.: Heterogeneous reaction of $Cl_2$ and $NO_2$ on γ-$Al_2O_3$: A potential formation pathway of secondary aerosols, Atmos. Environ., 188, 25-33, doi: 10.1016/j.atmosenv.2018.06.005, 2018.

Underwood, G.M., Miller, T.M., Grassian, V.H.: Transmission FT-IR and Knudsen Cell Study of the Heterogeneous Reactivity of Gaseous Nitrogen Dioxide on Mineral Oxide Particles, The Journal of Physical Chemistry A, 103, 6184-6190, doi: 10.1021/jp991586i, 1999.

Wang, L., Wang, W., Ge, M.: Heterogeneous uptake of $NO_2$ on soils under variable temperature and relative humidity conditions, J. Environ. Sci.-China, 24, 1759-1766, doi: org/10.1016/S1001-0742(11)61015-2, 2012.

WEDEPOHL, K.H.: The composition of the continental crust, Geochimicaet Cosmochimica Acta, 59, 1217-1232, doi: org/10.1016/0016-7037(95)00038-2, 1995.

Wu, L., Tong, S., Ge, M.: Heterogeneous Reaction of $NO_2$ on $Al_2O_3$: The Effect of Temperature on the Nitrite and Nitrate Formation, J. Phys. Chem. A, 117, 4937-4944, doi: 10.1021/jp402773c, 2013.

Wu, L.W., Tong, S.R., Wang, W.G., Ge, M.F.: Effects of temperature on the heterogeneous oxidation of sulfur dioxide by ozone on calcium carbonate, Atmos. Chem. Phys., 11, 6593-6605, doi:2011.

Yang, W., Chen, M., Xiao, W., Guo, Y., Ding, J., Zhang, L., He, H.: Molecular Insights into NO-Promoted Sulfate Formation on Model $TiO_2$ Nanoparticles with Different Exposed Facets, Environ. Sci. Technol., 52, 14110-14118, doi: 10.1021/acs.est.8b02688, 2018.

Ye, C., Gao, H., Zhang, N., Zhou, X.: Photolysis of Nitric Acid and Nitrate on Natural and Artificial Surfaces, Environ. Sci. Technol., 50, 3530-3536, doi: 10.1021/acs.est.5b05032, 2016.

Ye, C., Zhang, N., Gao, H., Zhou, X.: Photolysis of Particulate Nitrate as a Source of HONO and $NO_x$, Environ. Sci. Technol., 51, 6849-6856, doi: 10.1021/acs.est.7b00387, 2017.

Yi, C., Kwak, J.H., Peden, C.H.F., Wang, C., Szanyi, J.: Understanding Practical Catalysts Using a Surface Science Approach: The Importance of Strong Interaction between BaO and $Al_2O_3$ in $NO_x$ Storage Materials, The Journal of Physical Chemistry C, 111, 14942-14944, doi: 10.1021/jp0763376, 2007.

Yi, C., Szanyi, J.: BaO/$Al_2O_3$/NiAl(110) Model $NO_x$ Storage Materials: The Effect of BaO Film Thickness on the Amorphous-to-Crystalline $Ba(NO_3)_2$ Phase Transition, The Journal of Physical Chemistry C, 113, 716-723, doi: 10.1021/jp808766n, 2008.

Zhang, P., Lee, J., Kang, G., Li, Y., Yang, D., Pang, B., Zhang, Y.: Disparity of nitrate and nitrite in vivo in cancer villages as compared to other areas in Huai River Basin, China, Sci. Total Environ., 612, 966-974, doi: 10.1016/j.scitotenv.2017.08.245, 2018.

---

## Author Comment (AC6) · 2 Aug 2019

The comment was uploaded in the form of a supplement:
https://www.atmos-chem-phys-discuss.net/acp-2019-315/acp-2019-315-AC6-supplement.pdf